# Recovering mixtures of fast-diffusing states from short single-particle trajectories

Alec Heckert[1]*, Liza Dahal[1,2], Robert Tjian[2,3], Xavier Darzacq[1]*

[1]Department of Molecular and Cell Biology, Li Ka Shing Center for Biomedical and Health Sciences, University of California, Berkeley, Berkeley, United States; [2]CIRM Center of Excellence, University of California, Berkeley, Berkeley, United States; [3]Howard Hughes Medical Institute, Berkeley, United States

**Abstract** Single-particle tracking (SPT) directly measures the dynamics of proteins in living cells and is a powerful tool to dissect molecular mechanisms of cellular regulation. Interpretation of SPT with fast-diffusing proteins in mammalian cells, however, is complicated by technical limitations imposed by fast image acquisition. These limitations include short trajectory length due to photo-bleaching and shallow depth of field, high localization error due to the low photon budget imposed by short integration times, and cell-to-cell variability. To address these issues, we investigated methods inspired by Bayesian nonparametrics to infer distributions of state parameters from SPT data with short trajectories, variable localization precision, and absence of prior knowledge about the number of underlying states. We discuss the advantages and disadvantages of these approaches relative to other frameworks for SPT analysis.

## Editor's evaluation

This paper will be of interest to the cellular biologists who perform single-particle tracking experiments and develop new tracking methodologies. The authors investigate a new way of estimating an unknown number of diffusion states from short single-molecule trajectories. Ideas developed in the paper are likely to be used for further algorithm development. The authors give the users access to a repository on GitHub that contains comprehensive code that supports the paper.

*For correspondence:
alecheckert@gmail.com (AH);
darzacq@berkeley.edu (XD)

**Competing interest:** The authors declare that no competing interests exist.

## Introduction

Biological processes are driven by interactions between molecules. To understand the role of a molecular species in a process, a central challenge is to measure subpopulations of the molecule engaged in distinct interactions without perturbing the living system. Some interactions – such as complex formation – cause changes in a molecule's mobility. As a result, live-cell single-particle tracking (SPT), by separately observing the motion of individual molecules, is a promising tool to meet this challenge (*Shen et al., 2017*).

While SPT originally targeted proteins on cellular membranes, advances in the past two decades led to intracellular applications (*Barak and Webb, 1982*, *Ghosh and Webb, 1994*, *Kubitscheck et al., 2000*, *Goulian and Simon, 2000*). These include the use of stochastic labeling to isolate a single emitter's path (*Manley et al., 2008*), a principle that can be extended into intracellular settings with genetically encoded photoconvertible proteins (*Ando et al., 2002*, *Wiedenmann et al., 2004*) or cell-permeable dyes (*Grimm et al., 2015*, *Grimm et al., 2016*). Another advance is pulsed or 'stroboscopic' excitation, which reduces blur associated with fast-diffusing emitters (*Elf et al., 2007*).

**Figure 1.** Overview of sptPALM. (**A**) Schematic of experimental setup. An inclined illumination source is used in combination with a high-numerical aperture (NA) objective to resolve molecules in a thin slice in a cell. The excitation laser is pulsed to limit motion blur. Tracking yields a set of short trajectories (mean track length 3–5 frames). Trajectories shown are from a 7.48 ms tracking movie with retinoic acid receptor α-HaloTag (RARA-HaloTag) labeled with photoactivatable JF549 in U2OS nuclei. Asterisks in the movie frames mark particles at the edge of the focus. (**B**) Schematic of our inference problem. Each trajectory's state is assumed to be a random draw from a distribution of state parameters. The goal is to recover this distribution from the observed trajectories. (**C**) Effects of particle mobility on trajectory length. RARA-HaloTag trajectories from U2OS nuclei were binned into five groups based on their mean squared displacement (MSD). Individual data points are the mean trajectory length of each group for three distinct knock-in clones of RARA-HaloTag (c156: 36961 trajectories, c239: 27543 trajectories, c258: 60347 trajectories); bar heights are the means across clones.

The online version of this article includes the following figure supplement(s) for figure 1:

**Figure supplement 1.** Demonstrations of the effect of motion blur on the sptPALM measurement.

Together with modifications of TIRF microscopes (*Tokunaga et al., 2008*), these techniques have facilitated the application of SPT to intracellular settings with fast-moving subpopulations (*English et al., 2011*, *Persson et al., 2013*, *Izeddin et al., 2014*, *Normanno et al., 2015*, *Hansen et al., 2017*). Following *Manley et al., 2008*, we refer to this experiment as 'sptPALM' (*Figure 1A*, *Video 1*).

sptPALM experiments on fast-moving emitters in 3D settings pose several challenges for analysis (*Hansen et al., 2018*). First, apparent motion in sptPALM reflects both the true motion of the emitter and error associated with the estimate for its position ('localization error') (*Martin et al., 2002*, *Matsuoka et al., 2009*). Like fixed cell PALM and STORM microscopies (*Betzig et al., 2006*, *Rust et al., 2006*), the magnitude of localization error in sptPALM depends on the number of photons collected from each emitter (*Thompson et al., 2002*). But unlike fixed cell microscopies, sptPALM has another component of error due to *motion blur*, the convolution of the microscope's point spread function with the path of the emitter. This component of error is not trivial: the mean 2D displacement of a Brownian particle with diffusion coefficient 10 $\mu m^2 \, s^{-1}$ during a 1 ms integration is ~180 nm, substantially larger than typical localization error in fixed cell PALM/STORM (*Figure 1—figure supplement 1B*). Consequently, localization error in sptPALM depends on both the emitter's mobility and

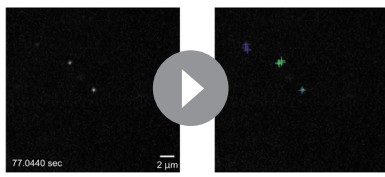

**Video 1.** Example of sptPALM data. NPM1-HaloTag in U2OS osteosarcoma nuclei was labeled with 100 nM PA-JFX549-HTL for 5 min followed by washes ('Materials and methods'), then imaged with a HiLo setup at 7.48 ms frame intervals with 1.5 ms excitation pulses. The pixel size after accounting for magnification is 160 nm. Dots and lines indicate the output of the detection and tracking algorithm; each trajectory has been given a distinct color.

https://elifesciences.org/articles/70169/figures#video1

**Video 2.** Illustration of defocalization for a single regular Brownian state. Trajectories were simulated in a 5 × 5 × 10 µm ellipsoid µm using the Euler–Maruyama scheme for regular Brownian motions with specular reflections at the ellipsoid boundaries. The diffusion coefficient for all trajectories was held constant at 2.0 µm² s⁻¹, while trajectories were randomly photoactivated at any point in the sphere and were subject to Poisson bleaching at 14 Hz. The left panel shows the 3D context of the trajectories, with dotted lines indicating the boundaries of the focal volume. The depth of the focal volume was 700 nm, which is roughly equivalent to the measured depth of field for our oil immersion objectives. The right panel shows the projection of the trajectories that coincide with the focal volume onto a hypothetical idealized camera. Notice that particles may make multiple transits through the focal volume that manifest as distinct trajectories.

https://elifesciences.org/articles/70169/figures#video2

its distance from the focus and is not simple to measure (*Kubitscheck et al., 2000*, *Berglund, 2010*, *Michalet and Berglund, 2012*). Pulsed excitation can be used to reduce motion blur (*Elf et al., 2007*), but because the laser pulse still has nonzero duration (usually ≥1 ms), motion blur remains an important part of the measurement (*Deschout et al., 2012*, *Lindén et al., 2017*).

Second, the high numerical aperture (NA) objectives required to resolve single emitters induce short depths of field, typically less than a micron. Whereas bacteria such as *Escherichia coli* are often small enough to fit into the resulting focal volume, mammalian cells – with depths ≥5–10 µm – cannot. As a result, intracellular SPT experiments only capture short transits of emitters through the focal volume, a behavior termed *defocalization* (*Figure 1C*, *Video 2*, *Video 3*; *Kues and Kubitscheck, 2002*, *Mazza et al., 2012*, *Hansen et al., 2018*). The duration of each transit depends on the emitter's mobility. This creates a sampling problem: slow particles with long residences inside the focal volume contribute a few long trajectories, while fast particles with short residences contribute many short trajectories. Mean trajectory length is often as little as 3–4 frames, severely limiting the ability to infer dynamic parameters (such as diffusion coefficient) from any single trajectory. Fast multifocal imaging may mitigate this problem (*Abrahamsson et al., 2013*), but such methods currently require higher photon budgets and are not yet applicable to fast-diffusing targets with high motion blur. Meanwhile, the use of cylindrical optics to encode axial position in PSF astigmatism (*Kao and Verkman, 1994*), while popular in fixed cell PALM/STORM, is complicated in sptPALM by its resemblance to motion blur.

Third, the true number of dynamic subpopulations or 'states' for a protein of interest is usually unknown a priori. Proteins often participate in many complexes with distinct dynamics. Model-dependent analyses that assume a fixed number of states (*Mazza et al., 2012*, *Hansen et al., 2017*, *Hansen et al., 2018*), while powerful when combined with complementary measurements (*Izeddin et al., 2014*, *Hansen et al., 2020*), are limited to measuring coefficients of known models. To compound model complexity, a protein may behave differently in distinct

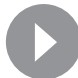

**Video 3.** Illustration of defocalization for multistate regular Brownian motion. Trajectories were drawn from two states – a fast state with diffusion coefficient 5.0 µm² and a slow state with diffusion coefficient 0.05 µm² s⁻¹ – and simulated with a spherical nucleus with 5 µm radius. The left panel shows the trajectories in their native three dimensions while the right panel shows trajectories projected through the focal volume.

https://elifesciences.org/articles/70169/figures#video3

subcellular environments. Indeed, although sptPALM directly observes the spatial context for each trajectory (*Xiang et al., 2020*), analyses such as jump distribution modeling often discard this information by aggregating jumps across all subcellular locations.

The central problem for sptPALM analysis is to recover the underlying dynamic states for a protein of interest given a set of observed trajectories in the presence of these three challenges.

A common approach to recover subpopulations from sptPALM is to construct histograms of the mean squared displacement (MSD), the maximum likelihood estimator for the diffusion coefficient in the absence of localization error. The MSD is highly variable for short trajectories and, when used to estimate diffusion coefficient, becomes especially error-prone when the variance of localization error is unknown (*Michalet and Berglund, 2012*). More problematically, MSD histograms assume that sampling from slow and fast states with equal occupation produces the same number of trajectories, which leads to severe state biases in the presence of defocalization (*Mazza et al., 2012*, *Hansen et al., 2018*). Common preprocessing steps to select for long trajectories compound the problem by introducing biases for slow emitters that remain in focus.

Methods based on least-squares fitting of the jump length cumulative distribution function (CDF) have interpreted sptPALM data with two- and three-state models while accounting for defocalization (*Mazza et al., 2012*, *Hansen et al., 2018*), but extend poorly to more complex models due to overfitting and do not provide a way to select between competing models.

A different approach to model selection is represented by vbSPT, a variational Bayesian framework for reaction-diffusion models (*Persson et al., 2013*). vbSPT relies on the evidence lower bound to identify the number of states, and it excels at recovering occupations and transition rates for a small number of diffusing states from short trajectories. However, it is not appropriate to apply in situations where the target's dynamic profile is not discrete and does not consider defocalization or localization error, although it can be complemented with a separate estimate of localization error (*Lindén et al., 2017*). As such, there is a need for methods that combine the advantages of Bayesian methods like vbSPT with a model that can accommodate nondiscrete dynamic profiles, while accounting for biases induced by sptPALM imaging geometry.

Here, we examine two alternative methods for recovering an sptPALM target's dynamic profile. The first is based on a Dirichlet process mixture model (DPMM) and the second on a finite state approximation to the DPMM that we refer to as a state array (SA). Exploring these techniques on simulated and real datasets, we find that although both DPMMs and SAs recover complex mixtures of states and can be applied to nondiscrete distributions of diffusion coefficients, SAs far outperform DPMMs due to their robustness to variable localization error variance. Both methods share the limitation that they do not deal with transitions between states. We investigate how this limitation affects apparent state occupations recovered with these methods.

The SA method is publicly available as the `pip`-installable Python package `saspt` (source: https://github.com/alecheckert/saspt; *Heckert, 2022c* documentation: https://saspt.readthedocs.io/en/latest/).

## Results
### Two approaches to infer subpopulations in sptPALM datasets

We considered how to infer dynamic subpopulations from the short, fragmented trajectories produced by sptPALM in a manner robust to the effects of localization error and defocalization (*Figure 1*).

A simple and popular approach to this problem is to make a separate estimate for the parameters of each trajectory, then compile a histogram of the results. In the case of Brownian motion, we refer to this method as the 'MSD histogram' approach since the MSD is the maximum likelihood estimator for the diffusion coefficient of a Brownian motion with no localization error.

Real estimates of a particle's position, however, are invariably associated with localization error. In sptPALM, this problem is more significant due to motion blur, which increases the magnitude of the error (*Figure 1—figure supplement 1*). To incorporate these effects, we refer to the combination of regular Brownian motion with normally distributed, mean-zero localization error as 'RBME' ('Materials and methods'). Each RBME is characterized by two parameters: the diffusion coefficient and the localization error variance. (For brevity, we refer to the latter simply as 'localization error.') Importantly,

the increments of RBME are only Markovian when the localization error is zero (*Martin et al., 2002*; *Figure 1—figure supplement 1*).

Because individual trajectories produced by sptPALM are usually too short to estimate localization error, and because it does not take into account other effects like defocalization, the MSD histogram approach is prone to large systematic biases (*Michalet and Berglund, 2012*, *Hansen et al., 2018*). While techniques exist to mitigate some biases of MSD fitting (*Kepten et al., 2015*), most are difficult to apply at the single trajectory level due to the small number of points per trajectory.

A distinct approach is represented by Bayesian finite state mixture models (*Marin et al., 2005*, *McLachlan et al., 2019*; *Figure 2A*, *Figure 2—figure supplement 1A*). Such models are comprised of a collection of *states* labeled $k = 1, ..., K$. Each state is associated with an occupation $\tau_k$ (describing the probability to observe trajectories from that state) and a vector of state parameters $\boldsymbol{\theta}_k$ (describing the kind of trajectories produced by that state). Importantly, $\boldsymbol{\theta}_k$ can also incorporate measurement parameters like the localization error. The probability to observe a particular trajectory $x$ is then $\sum_{k=1}^{K} \tau_k p_X(x|\boldsymbol{\theta}_k)$, where $p_X(x|\boldsymbol{\theta}_k)$ is a distribution over trajectories produced by state $k$ and depends on the type of motion being considered. The goal is to infer $\tau_k$ and $\boldsymbol{\theta}_k$ for each state given some observed set of trajectories $\boldsymbol{X}$. A challenge with such methods is choosing the number of states $K$ as well as the high computational cost when $p_X(x|\boldsymbol{\theta})$ is nonconjugate to the prior over $\boldsymbol{\theta}$.

Potential solutions can be found in the Bayesian nonparametric class of methods. These approaches begin with a single model comprising a very large or infinite collection of states. A Bayesian inference algorithm is then used to prune away superfluous complexity, leaving a sparse subset of states sufficient to explain the observed trajectories. The foundational example is the DPMM (*Ferguson, 1973*), which has the distinct advantage of being able to approximate essentially any mixture of states, discrete or continuous (*Neal, 1992*, *Teh, 2010*; *Figure 2B*). Its disadvantage is the high computational cost associated with inference, which becomes especially severe when considering types of motion with multiple parameters (such as RBME) (*Neal, 2000*, *Andrieu et al., 2003*).

We considered two responses to this challenge. First, we constructed a DPMM that uses a cheap approximation to RBME by treating the RBME as a Markov process (*Matsuda et al., 2018*; *Figure 3C*). This assumption is strictly true only when the localization error is zero and is the same assumption used to estimate diffusion coefficient via the MSD (*Michalet and Berglund, 2012*). Because localization error is never actually zero, we were curious to see when and how this method breaks down.

The second approach we explored is a model we refer to as a 'state array' (SA). This model is a special case of the finite state mixture, obtained by selecting a large number of states $K$ and fixing the state parameters to the vertices of an 'array' that spans some target parameter space (*Figure 2C*, *Figure 2—figure supplement 1*). For example, the array for RBME might span a range of biologically plausible diffusion coefficients and localization error variances. An array for an anomalous diffusion model may also incorporate one or more anomaly parameters. The occupation of each 'state' in this array is inferred through a variational Bayesian algorithm, driving the occupation of most states to zero to leave a minimal set sufficient to explain the observations ('Materials and methods'). Importantly, SAs jointly infer a 'global' distribution over the state parameters along with 'individual' distributions for each trajectory. The nature of the variational inference algorithm means that the 'global' distribution is always a weighted mean of these 'individual' distributions. We focus our attention on the global distribution in this article, with some consideration of the individual distributions for each trajectory at the end.

Because the parameters for each state in an SA are fixed, the most expensive computations can be cached and reused throughout inference. As a result, SAs can handle more complex models than DPMMs. In this article, we use a 2D SA for RBME spanning a range of diffusion coefficients and localization error variances. After inference, we marginalize out the localization error part to yield 1D functions of the diffusion coefficient (*Figure 3B*). This procedure naturally incorporates uncertainty about localization error variance, rendering SAs more robust to variations in localization error than DPMMs (*Figure 3—figure supplement 1*).

DPMMs and SAs work best with thousands to tens of thousands of trajectories. This often requires aggregating trajectories across multiple cells, which can mask cell-to-cell variability. To assess cell-to-cell variability, we also found it useful to have a 'cheap and dirty' estimate of state occupation that works with a smaller number (100 s) of trajectories. This is derived from the SA calculation and is simply the sum of the normalized RBME likelihood function across all of the trajectories observed in

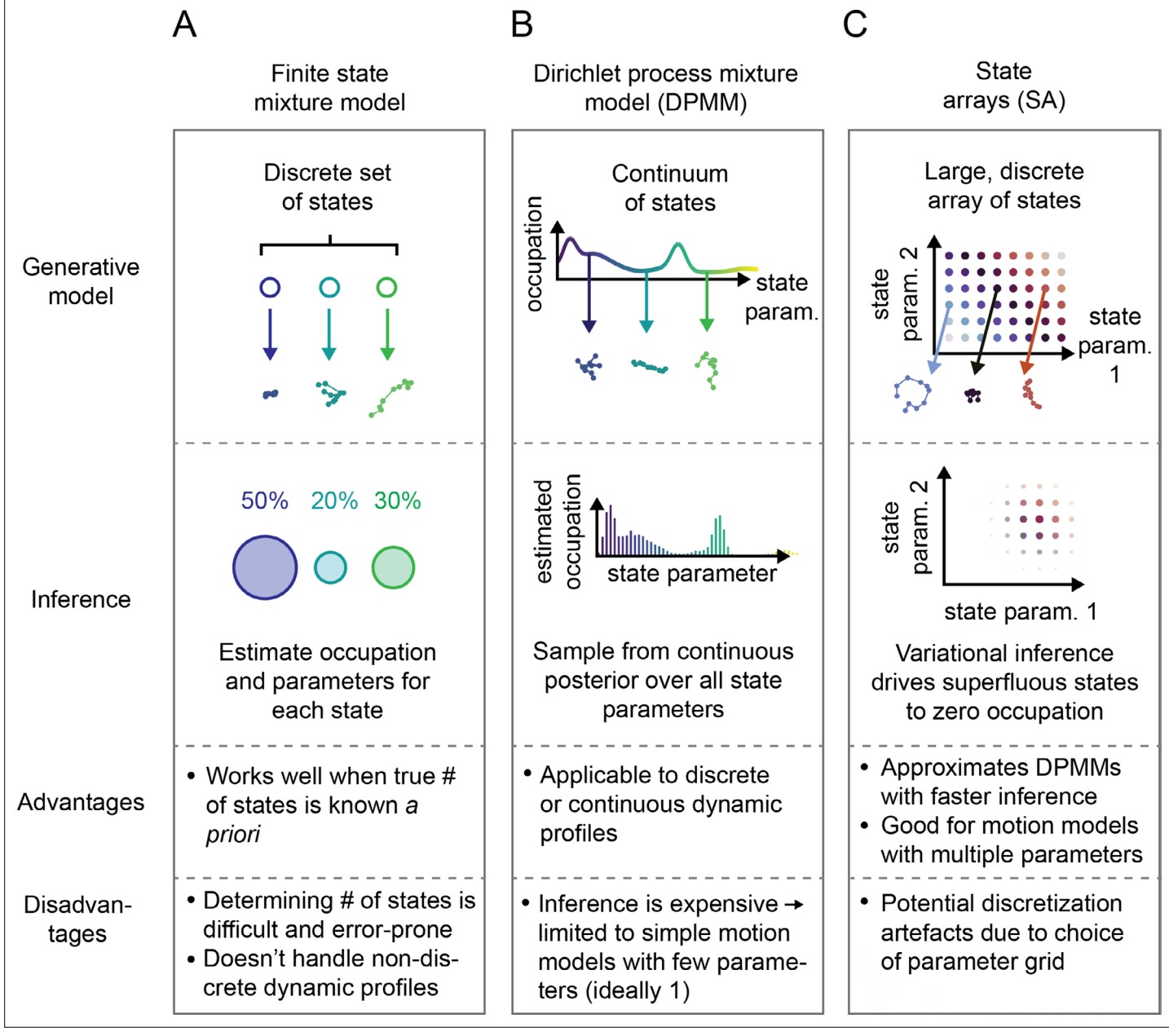

**Figure 2.** Schematic comparison of finite state mixtures, Dirichlet process mixtures, and state arrays (SAs). (**A**) Finite state mixture models use a discrete set of $K$ states. Challenges include estimating $K$ and producing intelligible output when the underlying dynamic profile is not discrete. (**B**) Dirichlet process mixture models (DPMMs) address the problem of nondiscrete dynamic profiles by using a continuous distribution over state parameters. Inference routines are slow, so in this work we use approximative motion models. (**C**) SAs, a special case of the finite state mixture. SAs approximate DPMMs by using a discrete grid of state parameters and have a faster inference routine. Challenges with SAs include the choice of the parameter grid.

The online version of this article includes the following figure supplement(s) for figure 2:

**Figure supplement 1.** Probabilistic graphical models for finite state mixtures, Dirichlet process mixtures, and state arrays.

a cell. We refer to this as the 'naive occupation estimate.' Functionally it behaves like a less precise version of the SA method ('Materials and methods').

Finally, to account for defocalization we developed a method applicable to the posterior distributions of both DPMMs and SAs (*Figure 3—figure supplement 2*, 'Materials and methods').

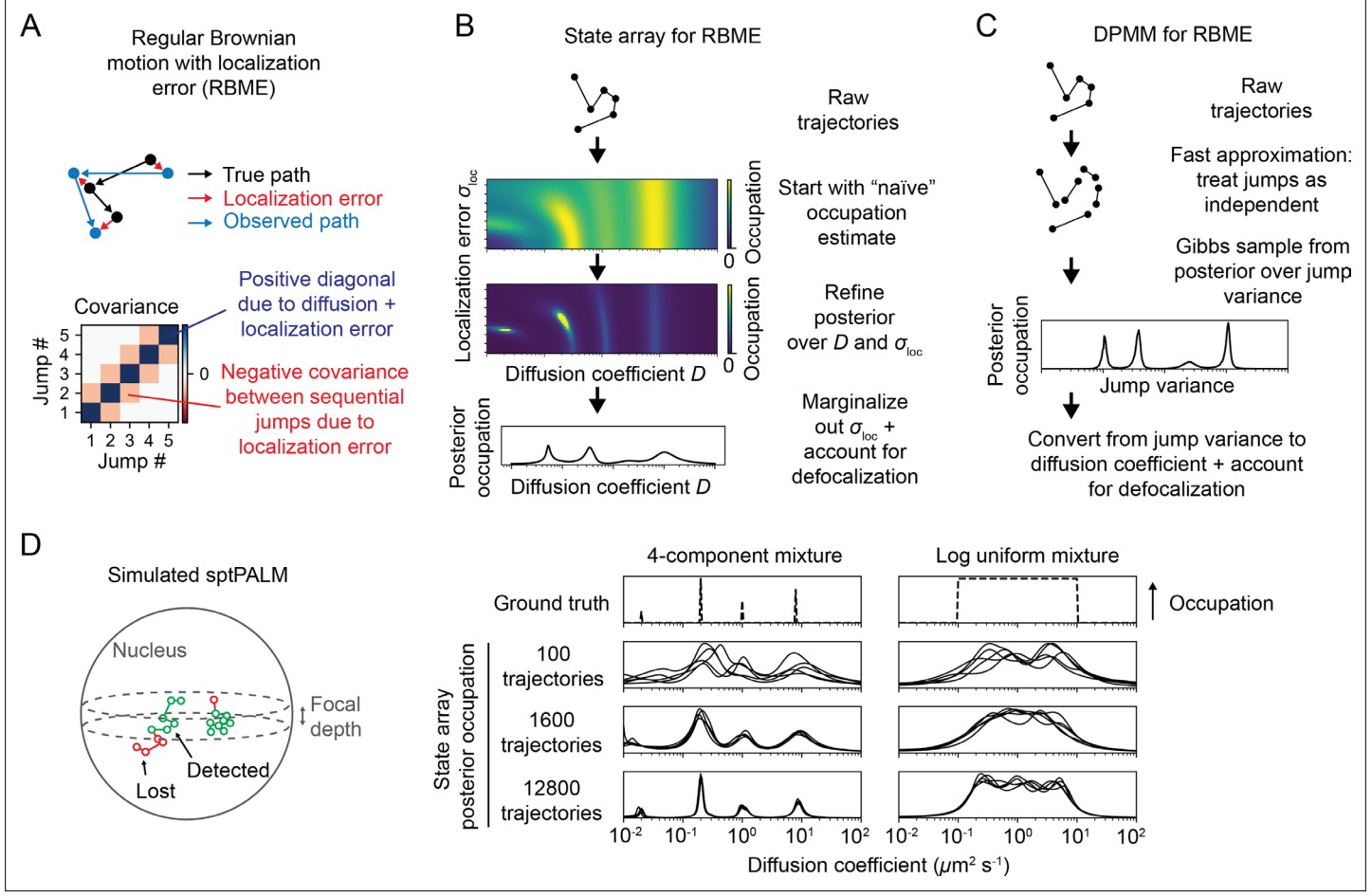

**Figure 3.** Application of state arrays and Dirichlet process mixture models (DPMMs) to mixtures of Brownian motions. (**A**) Regular Brownian motion with localization error (RBME) is a motion model that involves two parameters: diffusion coefficient and localization error variance. (For brevity, we refer to the latter simply as 'localization error.') Unlike pure Brownian motion, RBME has correlations between sequential jumps due to the influence of localization error. (**B**) State array inference for RBMEs. The naive occupation estimate is the initial estimate for the posterior, which is subsequently refined through variational inference. At the end of inference, we marginalize out localization error to yield 1D distributions over the diffusion coefficient. (**C**) DPMM inference for mixtures of Brownian motions. Because the Gibbs sampling routine for a pure DPMM is slow, we use an approximative motion model that neglects the off-diagonal terms of the covariance matrix in (**A**). (**D**) Example of state arrays evaluated on simulated sptPALM. Tracking was simulated in a spherical nucleus with 700 nm focal depth, uniform photoactivation probability, 14 Hz bleaching rate, 7.48 ms frame intervals, and variable localization error. The lines represent the state array posterior mean occupations for independent replicates of the same simulation.

The online version of this article includes the following figure supplement(s) for figure 3:

**Figure supplement 1.** Challenges distinguishing between diffusion and localization error.

**Figure supplement 2.** Accounting for the influence of defocalization on state occupations.

## Evaluating DPMMs and SAs on simulated sptPALM data

As the target for inference, we considered a mixture of RBMEs enclosed in a spherical membrane with a thin focal volume bisecting the sphere, with dimensions similar to a mammalian cell nucleus. Emitters photoactivate and photobleach throughout the sphere and are only observed when their positions coincide with the focal volume. Because no gaps are allowed during tracking, the result is a highly fragmented set of trajectories with mean length 3–5 frames. We chose simulation settings to approximate real sptPALM experiments, with bleaching rates ≥10 Hz, diffusion coefficients in the range 0–100 $\mu m^2\ s^{-1}$, and localization error variances between $0^2$ and $0.06^2 \mu m^2$.

We compared the ability of DPMMs, SAs, and MSD histograms to recover the underlying distribution of diffusion coefficients from this data. We divided these simulations into four classes with increasing difficulty. In class 1, localization error for all states was provided as a known constant to the algorithms (*Figure 4A*, *Figure 4—figure supplement 1A*). In class 2, localization error was held

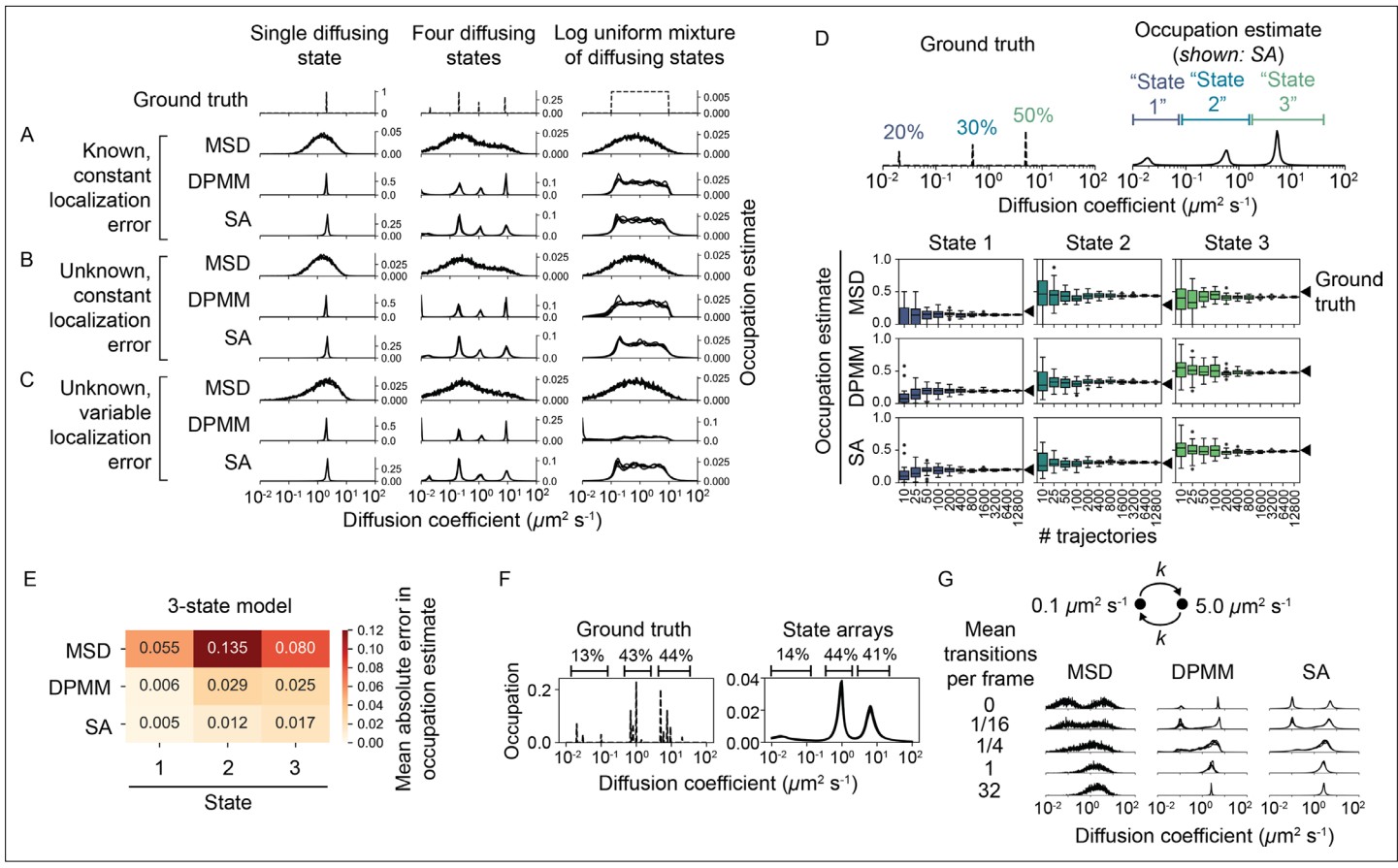

**Figure 4.** Comparison of the mean squared displacement (MSD) histogram, Dirichlet process mixture model (DPMM), and state array (SA) methods to recover dynamic profiles from trajectory simulations. (**A–C**) Mixtures of diffusing states were simulated in a 700 nm focal volume with 7.48 ms frame intervals. Simulations were divided into three classes of increasing difficulty based on the treatment of localization error as described in the text. For each replicate, exactly 12,800 trajectories were simulated. Estimated occupations for five independent replicates are overlaid on each subplot. (**D**) Accuracy of state occupation estimates for each method as a function of sample size. Each method was run on trajectory simulations generated from an underlying three-state dynamic model (0.02 $\mu$m$^2$ s$^{-1}$ [20%], 0.5 $\mu$m$^2$ s$^{-1}$ [30%], 5.0 $\mu$m$^2$ s$^{-1}$ [50%]), then occupations were estimated by integrating the distribution produced by each method. Limits of integration were set to 0–0.08 $\mu$m$^2$ s$^{-1}$ (state 1), 0.08–1.5 $\mu$m$^2$ s$^{-1}$ (state 2), or 1.5–40 $\mu$m$^2$ s$^{-1}$ (state 3). 20 replicates were run per condition. (**E**) Mean absolute error (MAE) in state occupation estimates for the simulations in (**D**). Each value is the average MAE across all replicates. (**F**) Inferring mixtures of diffusing states with similar diffusion coefficients using SAs. For each replicate, a total of 6400 trajectories were simulated with the indicated underlying state distribution. (**G**) Effect of state transitions on the MSD, DPMM, and SA approaches. We varied the first-order transition rate constant between two diffusing states, simulating 6400 trajectories per replicate.

The online version of this article includes the following figure supplement(s) for figure 4:

**Figure supplement 1.** Comparison of Dirichlet process mixture models (DPMMs), state arrays (SAs), and mean squared displacement (MSD) histograms to estimate state occupations for several kinds of trajectory simulation.

**Figure supplement 2.** Quantitative comparison of the results in *Figure 4—figure supplement 1*.

**Figure supplement 3.** Effect of sample size on accuracy and precision of the mean squared displacement (MSD) histogram, Dirichlet process mixture model (DPMM), and state array (SA) methods using two-state trajectory simulations.

**Figure supplement 4.** Effect of sample size on accuracy and precision of the mean squared displacement (MSD) histogram, Dirichlet process mixture model (DPMM), and state array (SA) methods using four-state trajectory simulations.

**Figure supplement 5.** State array and Dirichlet process mixture model (DPMM) performance on optical–dynamical simulations of sptPALM movies with two diffusing states.

**Figure supplement 6.** State array and Dirichlet process mixture model (DPMM) accuracy 2.

**Figure supplement 7.** Effect of state transitions on Dirichlet process mixture model (DPMM) and state array (SA) methods.

**Figure supplement 8.** Performance of Dirichlet process mixture model (DPMM) and state array (SA) methods on states with diffusion coefficients slower than the minimum diffusion coefficient included in the support.

*Figure 4 continued on next page*

*Figure 4 continued*

**Figure supplement 9.** Performance of Dirichlet process mixture model (DPMM) and state array (SA) algorithms on clusters of states with similar diffusion coefficients.

**Figure supplement 10.** Comparison of state arrays with vbSPT with optical–dynamical simulations.

**Figure supplement 11.** State arrays applied to fractional Brownian motion with localization error (FBME) using optical–dynamical simulations.

**Figure supplement 12.** Systematic errors in fractional Brownian motion with localization error (FBME) parameter retrieval due to motion blur.

---

constant for all states but was unknown to the algorithms (*Figure 4B*, *Figure 4—figure supplement 1B*). In class 3, localization error was allowed to vary between diffusive states and was also unknown to the algorithms (*Figure 4C*, *Figure 4—figure supplement 1C*). Finally, for class 4 we simulated full sptPALM-like movies that incorporate heterogeneous localization error, motion blur, camera noise, tracking errors, and defocus (*Figure 4—figure supplement 5*, *Figure 4—figure supplement 6*). In these simulations, the localization error is unique for each emitter and depends on the emitter's axial position, the stochastic number of photons it emits during each integration, and its pattern of motion blur (*Video 4*, *Video 5*).

DPMMs and SAs both recovered the dynamic profile for simulations in class 1 with a resolution that exceeded the MSD histogram approach. With large samples of trajectories, DPMMs and SAs inferred even nondiscrete distributions of states (*Figure 4A*, *Figure 4—figure supplement 1A*).

When knowledge of the localization error was removed (classes 2 and 3), the SA approach outperformed both the MSD and DPMM approaches. The DPMM's performance was especially poor when the contributions of diffusion and error to jump variance were similar ($D\Delta t \approx \sigma_{\text{loc}}^2$), likely due to its simplistic treatment of localization error. Meanwhile, the dynamic profile estimated by SAs was unperturbed by variations in the localization error (*Figure 4B and C*, *Figure 4—figure supplement 1B and C*). Comparing the results from simulations in class 3 numerically, we found that the root mean squared deviation of the estimated CDF from the true CDF was ≤ 5% for SAs, while it was 5–20% for both the MSD histogram and DPMM approaches (*Figure 4—figure supplement 2*).

The dynamic profiles produced by the MSD, DPMM, and SA approaches can be integrated to yield occupation estimates over particular diffusion coefficient ranges. We compared the accuracy and precision of these estimates with discrete two-, three-, or four-state models (*Figure 4D*, *Figure 4—figure supplement 3*, *Figure 4—figure supplement 4*). As the number of trajectories increased, occupations estimated by DPMMs and SAs converged to within 3% of the true values. In contrast, the MSD approach was associated with large systematic errors, an effect previously reported (*Mazza et al., 2012*, *Hansen et al., 2018*).

On full optical and dynamic simulations in class 4, SAs also outperformed the DPMM approach

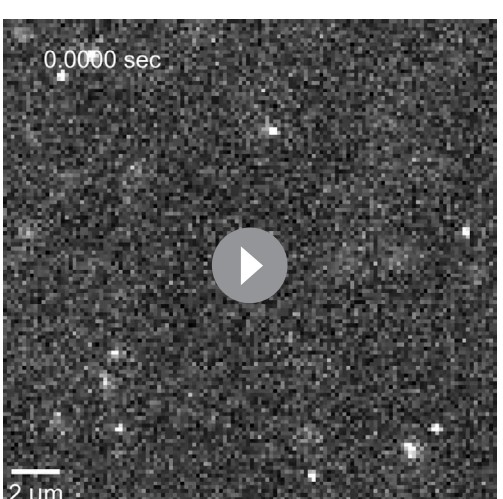

**Video 4.** Example of a simulated SPT movie. Two Brownian states with diffusion coefficients 0.01 and 5.0 µm² s⁻¹ were simulated; imaging was simulated with settings similar to our experimental SPT system including an objective with numerical aperture 1.49, immersion medium with refractive index 1.515, image pixel size 0.16 µm, frame interval 7.48 ms, and 2 ms excitation pulses. Simulations were performed using the sptPALMsim package.
https://elifesciences.org/articles/70169/figures#video4

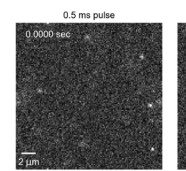
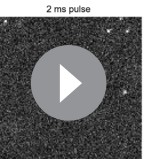
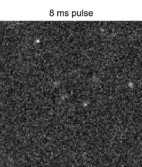

**Video 5.** Simulated SPT movies at variable excitation pulse widths. Mixtures of Brownian motions were simulated as in *Video 4*, except we used 0.5, 2.0, or 8.0 ms excitation pulses and a frame interval of 20 ms. The mixture had four diffusing states with the following diffusion coefficients: 0.1 µm² s⁻¹ (10% occupation), 2.5 µm² s⁻¹ (20% occupation), 9.0 µm² s⁻¹ (30% occupation), and 20.0 µm² s⁻¹ (40% occupation).
https://elifesciences.org/articles/70169/figures#video5

(*Figure 4—figure supplement 5*, *Figure 4—figure supplement 6*). Again, the difference was particularly pronounced for small diffusion coefficients, for which the DPMM state occupation estimates were severely inaccurate. Both methods had difficulty recovering the fastest diffusion coefficient tested (*Figure 4—figure supplement 5B*), possibly due to the restrictive conditions on the maximum jump distance used during tracking.

A central limitation of DPMMs and SAs is that they do not account for transitions between diffusive states. To determine the effect of state transitions on the output of these algorithms, we simulated mixtures of two diffusive states with increasing transition rates (*Figure 4G*, *Figure 4—figure supplement 7*). While slow transition rates had a negligible effect on the estimated state profile, transition rates approaching the frame interval appeared as single state with intermediate diffusion coefficient (*Figure 4—figure supplement 7C*), consistent with a result from reaction-diffusion systems (*Crank, 1975*). The shift from the two-state to single-state regime occurred in a narrow window of mean state dwell times between 0.05 and 0.5 frame intervals.

In this article, we restricted DPMM/SA inference to a range of diffusion coefficients from $10^{-2}$ to $10^2$ $\mu m^2$ $s^{-1}$. We also explored what happens when the true diffusion coefficient lies outside this range. DPMMs and SAs still recovered the correct state occupations by using the closest diffusion coefficient in their respective supports (*Figure 4—figure supplement 8*).

In the presence of multiple diffusing states with similar diffusion coefficients, both DPMMs and SAs tended to identify a single population with occupation equal to the sum of the occupations for each true state (*Figure 4F*, *Figure 4—figure supplement 9*).

We compared the performance of SAs and vbSPT (*Persson et al., 2013*) using simulated SPT movies with different dynamic models (*Figure 4—figure supplement 10*). Both methods had comparable accuracy on simple two-state models (*Figure 4—figure supplement 10B*). On more complex models (*Figure 4—figure supplement 10C*), both methods encountered distinct difficulties, with vbSPT tending to overestimate and SAs tending to underestimate the number of states. For clusters of states with similar parameters (*Figure 4—figure supplement 10C*, bottom), SAs tend to produce a 'smear' of state occupations over a range of diffusion coefficients, while vbSPT tended to produce a different cluster of states in the same region of parameter space. vbSPT was noticeably less accurate at recovering slow-moving states with small diffusion coefficients ($<0.1$ $\mu m^2$ $s^{-1}$). We concluded that both approaches are useful and may provide complementary information.

While our investigation focused primarily on Brownian motion, SAs can be applied to any motion model parameterized by a likelihood function. To explore applications of SAs outside of Brownian motion, we applied it to fractional Brownian motion (FBM), a generalization of Brownian motion capable of producing anomalous diffusion (*Mandelbrot and Van Ness, 1968*). Whereas Brownian motion's sole parameter is the diffusion coefficient, FBM parameterizes both the magnitude (via a scaling coefficient) and the temporal correlations (via the Hurst parameter) of a particle's increments. As with Brownian motion, we simulated sptPALM movies with fraction Brownian particles with variable diffusion coefficient and Hurst parameter (*Video 6*). To construct a state array for FBM, we used a 3D array over scaling coefficient, Hurst parameter, and localization error variance (*Figure 4—figure supplement 11C*). As with the RBME array, we marginalized out localization error after inference. While the SA accurately recovered the diffusion coefficient and Hurst parameter for multistate FBM models (*Figure 4—figure supplement 11D*), we noted a systematic error in the estimation of low (subdiffusive) Hurst parameters due to motion blur (*Figure 4—figure supplement 12*).

## Performance of state arrays on experimental sptPALM

After observing that SAs outperformed DPMMs on simulations, we proceeded to evaluate SAs on real data. We acquired an sptPALM dataset in U2OS osteosarcoma nuclei with endogenously tagged retinoic acid receptor-α-HaloTag

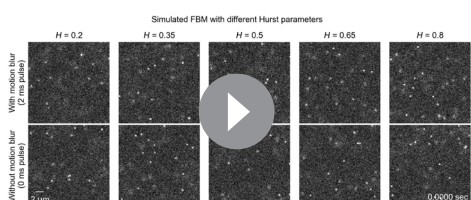

**Video 6.** Simulated SPT movies with fractional Brownian motion (FBM). FBMs with different Hurst parameters were simulated under conditions similar to *Video 4* with either 0 ms (instantaneous) or 2 ms excitation pulses and a frame interval of 7.48 ms. The scaling coefficients were modified to maintain the same jump variance between frames for all of the motions ('Materials and methods').

https://elifesciences.org/articles/70169/figures#video6

(RARA-HT) (*Pontén and Saksela, 1967*, *Los et al., 2008*; (*Figure 5—figure supplement 1*). RARA-HT is a type II nuclear receptor that heterodimerizes via its ligand-binding domain (LBD) with the retinoid X receptor (RXR) to form a complex competent to bind chromatin and regulate target genes *Giguere et al., 1987*, *Petkovich et al., 1987*, *Brand et al., 1988*, *Yu et al., 1991*, *Bugge et al., 1992*, *Marks et al., 1992*, *Leid et al., 1992*; reviewed in *Evans and Mangelsdorf, 2014*). In addition, association of coregulator complexes with the RAR/RXR heterodimer has been shown to influence the dimer's dynamics in FCS studies (*Brazda et al., 2011*, *Brazda et al., 2014*). As such, RARA-HT is expected to inhabit a variety of dynamic states in sptPALM.

For comparison, we also performed identical sptPALM experiments with histone H2B-HaloTag (H2B-HT), a protein with a high-occupation immobile state (*Hansen et al., 2017*, *McSwiggen et al., 2019*), as well as HaloTag and HaloTag-NLS (HT and HT-NLS), which are fast-diffusing proteins with low immobile fractions.

The four proteins presented distinct dynamic profiles (*Figure 5A*). For both HT and HT-NLS, the SA identified a single highly mobile state. In agreement with previous reports (*Xiang et al., 2020*), we observed that addition of the NLS reduces HaloTag's diffusion coefficient by two- to threefold. In contrast, both RARA-HT and H2B-HT had substantial immobile fractions, accounting for roughly 40 and 70% of their total populations, respectively (*Figure 5C*). SAs identified stark differences in the mobile subpopulations for RARA-HT and H2B-HT. Whereas H2B-HT presented a fast population at 8–10 $\mu m^2$ $s^{-1}$, RARA-HT inhabited a broad spectrum of diffusing states ranging from 0.3 to 10.0 $\mu m^2$ $s^{-1}$. Biological replicates gave similar results (*Figure 5—figure supplement 2A*).

To determine the origins of the dynamic states observed for RARA-HT, we performed domain deletions (*Figure 5B*). Removal of either the DNA-binding domain (DBD) or LBD resulted in loss of the immobile population. Because both the DBD and LBD are required for chromatin binding by the RAR/RXR heterodimer, this suggests that the immobile fraction represents chromatin-bound molecules. To confirm this, we introduced a point mutation (C88G) in the zinc fingers for the RARA-HT DBD that abolishes DNA-binding in vitro (*Zhu et al., 1999*). This led to loss of the immobile fraction (*Figure 5B*). Deletion of the unstructured N-terminal domain (NTD) or C-terminal domain (CTD) had a milder effect, suggesting that these domains are not the primary determinants of the dynamic behavior of RARA-HT.

To understand the origins of heterogeneity in the diffusive profile, we performed three variants of bootstrap aggregation (*Figure 5—figure supplement 2B*). The primary origins of variability for both DPMMs and SAs were cell-to-cell rather than clone-to-clone variability or intrinsic variability due to finite sample sizes.

## Spatiotemporal context of cellular protein dynamics

In the process of inferring the global distribution over state parameters for an sptPALM dataset, SAs jointly infer individual distributions for each trajectory. Up to this point, we have analyzed the global distribution. However, it is also possible to aggregate the individual distribution for each trajectory as a function of space or time, yielding, for instance, separate dynamic profiles for every spatial location in an experiment. This approach offers a potential route to understand spatiotemporal variation in the dynamics of a protein target.

We explored this aspect of SAs with a U2OS nucleophosmin-HaloTag (NPM1-HT) sptPALM dataset. NPM1-HT exhibits partial nucleolar localization (*Figure 6—figure supplement 1B*) and distinct dynamic behavior inside and outside nucleoli (*Mitrea et al., 2018*). The SA identified a broad range of diffusion coefficients for NPM1-HT, with three modes including an effectively immobile population (*Figure 6A*). Selecting four ranges of diffusion coefficients for analysis (*Figure 6A*), we visualized the posterior distribution as a function of space, calculating local fractional occupations for each range (*Figure 6B*, *Figure 6—figure supplement 1C*). This analysis revealed that some populations (including a slow-moving mobile population at 0.23 $\mu m^2$ $s^{-1}$) are enriched in nucleoli, while others (for instance, a fast-moving population at 4 $\mu m^2$ $s^{-1}$) are depleted and still others show no preference (*Figure 6C*). Notably, these preferences are apparent even in the naive occupations for trajectories in each compartment (*Figure 6—figure supplement 1D*).

The NPM1-HT tracking experiments were performed with an acquisition sequence comprising several phases with distinct levels of photoactivation. As a result, the localization density varied temporally in each movie. To understand the effect of localization density on the diffusion coefficient

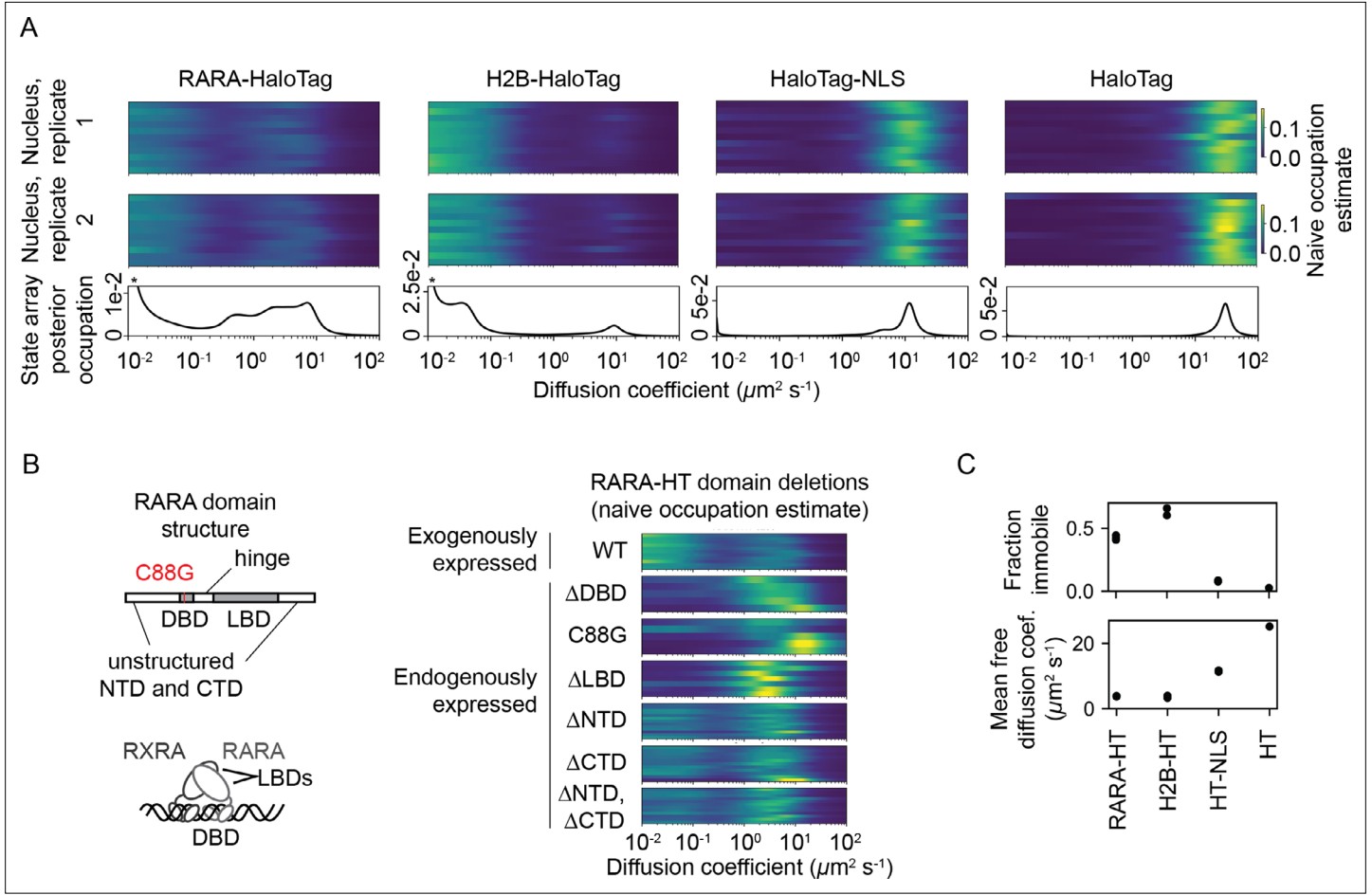

**Figure 5.** State arrays (SAs) applied to experimental sptPALM. All sptPALM experiments were performed with the photoactivatable dye PA-JFX549 using a TIRF microscope with HiLo illumination, 7.48 ms frame intervals, and 1 ms excitation pulses. (**A**) Naive and SA occupations for four different tracking targets. The upper two panels are the naive occupations for each nucleus in each of two biological replicates. Biological replicates correspond to separate knock-in clones for RARA-HaloTag or separate transfections for the other constructs (mean 1627 trajectories per nucleus). The bottom panel displays the SA occupations for a run of the SA algorithm on trajectories pooled from a single biological replicate (mean 17,899 trajectories per biological replicate). Asterisks for RARA-HaloTag and H2B-HaloTag indicate that the immobile fraction for these constructs has been truncated to visualize the faster-moving states. (**B**) Naive occupation estimate for RARA-HaloTag constructs bearing domain deletions or point mutations. 'Exogenously expressed' constructs were expressed from a nucleofected PiggyBac vector under an L30 promoter. (**C**) Quantification of the immobile fractions and mean free diffusion coefficients for the four constructs in (**A**). The 'immobile fraction' was defined as the total occupation below 0.05 μm² s⁻¹, while the mean free diffusion coefficient was the posterior mean diffusion coefficient above this threshold. Each dot represents a biological replicate (a different knock-in clone for RARA-HT or a different nucleofection for H2B-HT, HT-NLS, and HT).

The online version of this article includes the following source data and figure supplement(s) for figure 5:

**Source data 1.** Raw and labeled RARA-HaloTag Western blots used in *Figure 5*.

**Figure supplement 1.** Validation of endogenously tagged U2OS RARA-HaloTag cell lines.

**Figure supplement 2.** Assessing variability of state arrays (SAs) on experimental sptPALM by subsampling.

---

likelihoods, we aggregated the naive state occupations over 100-frame temporal blocks (*Figure 6D*). These experiments demonstrated that high localization densities led to a deflation in the occupation of slower-moving states, probably due to tracking errors. As a result, only phases with low localization density were used for posterior estimation. This demonstrates how the temporal perspective on the posterior may be useful as a guide for subsequent analysis, including quality control in SPT experiments.

## Discussion

Intracellular sptPALM with fast-diffusing proteins presents unique challenges for analysis. In particular, the issues of state bias arising from imaging geometry, limited information available from any

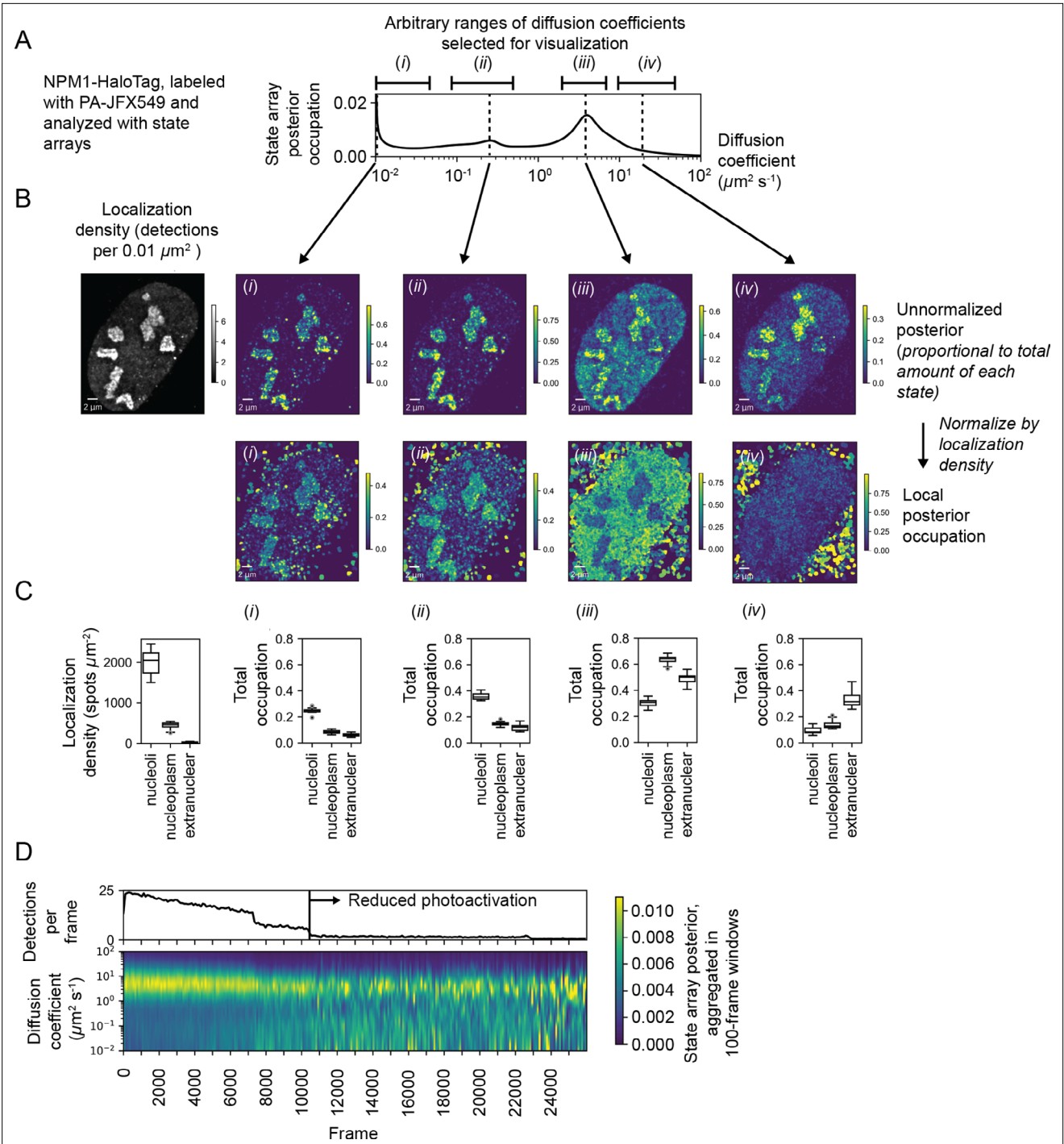

**Figure 6.** Spatiotemporal variation in the state array posterior distribution. (**A**) Posterior occupations for a state array evaluated on NPM1-HaloTag trajectories in U2OS nuclei. The ranges labeled i, ii, iii, and iv indicate parts of the dynamic profile isolated for analysis in subsequent panels. (**B**) Spatial distribution of the posterior probability in (**A**) for NPM1-HaloTag trajectories in a single U2OS nucleus. The posterior model over the diffusion coefficient was evaluated for each of the origin trajectories, and these points were then used to perform a kernel density estimate (KDE) with a 100 nm Gaussian kernel. For the local normalized occupation, these KDEs were normalized to estimate the relative fractions of molecules in each state. (**C**) Quantification of the analysis in (**B**) for 15 nuclei. 'Nucleoplasmic' trajectories were defined as trajectories outside nucleoli but inside the nucleus. (**D**) Temporal variation in the posterior distribution.

The online version of this article includes the following source data and figure supplement(s) for figure 6:

**Figure supplement 1.** Validation of U2OS NPM1-HaloTag lines.

**Figure supplement 1—source data 1.** Raw and labeled NPM1-HaloTag Western blots.

single trajectory, and variable localization error must be addressed prior to biological interpretation of sptPALM data.

The two methods investigated here, DPMMs and SAs, represent distinct approaches to this problem inspired by Bayesian nonparametrics. These methods identify sparse explanatory models from more complex alternatives, similar to other popular SPT approaches like vbSPT, but can use a broader range of dynamic models and are applicable when the dynamic profile is not comprised of discrete states. Between the two methods, SAs far outperformed DPMMs. By approximating continuous distributions over the diffusion coefficient with a grid of discrete states, SAs have qualitative similarities to recent methods to infer grids of dissociation rates from SMT trajectory length (*Reisser et al., 2020*).

When evaluated on real sptPALM data, SAs revealed previously unappreciated features of the dynamic profile for RARA-HaloTag and H2B-HaloTag. In particular, RARA-HaloTag exhibited a broad spectrum of diffusive states that stands in contrast to the more discretized profile of H2B-HaloTag or HaloTag-NLS. The ability to identify the presence or absence of discrete diffusing states is a major advantage of SAs over existing methods, which are generally premised on the existence of discrete states. We found that SAs were especially useful when complemented with the naive occupation estimate to visualize cell-to-cell and movie-to-movie variability. A Python tool that implements SAs can be found at https://github.com/alecheckert/saspt with documentation at https://saspt.readthe-docs.io.

DPMMs and SAs have several limitations. DPMMs require prior measurement of the localization error, while SAs require selection of a parameter grid with spacing fine enough to avoid discretization artifacts. The `saSPT` package uses default parameter grids that satisfy this requirement for regular and FBM. However, the grid needs to be reevaluated for any new types of motion to which SAs are applied. Additionally, neither DPMMs nor SAs consider transitions between states, a major short-coming of these methods.

Our experiments used a fixed range of diffusion coefficients from $10^{-2}$ to $10^2$ $\mu m^2$ $s^{-1}$. Even when the true diffusion coefficient was outside this range, SAs accurately estimated state occupations by using the nearest available diffusion coefficient (*Figure 4—figure supplement 8*). Our experimental SPT results, with large spikes at the lowest diffusion coefficient, suggest this is common in real data for SPT targets with very slow or immobile populations. A potential area for future improvement is to extend the support iteratively until the slowest and fastest states are captured. Such an approach would need to contend with the increased difficulty in estimating the diffusion coefficient when it is much smaller than the localization error variance (*Figure 3—figure supplement 1C*).

While we have only investigated the application of SAs to regular Brownian motion (and, briefly, FBM) in this article, the model could be extended to any type of motion parameterized by a likelihood function. We highlight two potential challenges for any such work. First, the SA's size scales with the number of parameters of the motion, meaning that more complex models are more computationally expensive. This could be addressed at the implementation level; for instance, by porting SA inference to graphical processing units. The second and more fundamental challenge is the similarity of the various flavors of anomalous diffusion to localization and tracking errors. For instance, both the Hurst parameter in FBM and the localization error primarily manifest as negative off-diagonal components of the trajectory increment covariance matrix (*Figure 4—figure supplement 11B*). Likewise, the erratic jumps of Levy flights have similarities to tracking errors. These issues are likely to become more significant when the sptPALM is lower in quality or highly heterogeneous (due to motion blur, defocus, and nonstationary camera noise).

In a recent objective evaluation of methods to measure anomalous diffusion (*Muñoz-Gil et al., 2021*), even top-performing methods (including recent machine learning approaches) were associated with mean absolute error $gt_{0.3}$ when estimating anomaly parameters for short trajectories (<10 frames). Because SAs create mixture models out of any underlying set of motion models, they could potentially be combined with such approaches (rather than the raw RBME likelihood function we use here) to boost their performance when run on large collections of short sptPALM trajectories.

Neither DPMMs nor SAs have any built-in mechanism to distinguish true jumps from tracking errors. Both rely on trajectories produced by another algorithm. It may be possible to combine both tracking and state occupation estimation into a single inference step using a model defining a joint distribution over states and possible links between detected particles.

## Materials and methods

### Plasmids

Unless otherwise noted, all PCRs were performed with New England Biosciences Phusion High-Fidelity DNA polymerase (M0530S), and Gibson assemblies (*Gibson et al., 2009*) were performed with New England Biosciences Gibson Assembly Master Mix (E2611S) following the manufacturer's instructions. Cloning and expression of plasmids was performed in *E. coli* DH5α using the Inoue protocol (*Im et al., 2011*). Plasmids used for nucleofections were purified with Zymo midiprep kit (Zymo D4200) and concentrations were quantified by absorption at 260 nm. Cloning primers were synthesized by Integrated DNA Technologies as 25 nmol DNA oligos with standard desalting, and sequences were verified by Sanger sequencing at the UC Berkeley DNA Sequencing Facility. A complete list of the primers used in this article is provided in *Supplementary file 1*, and a complete list of the plasmids used in this article is provided in *Supplementary file 2*.

We produced the vector PB PGKp-PuroR L30p MCS-GDGAGLIN-HaloTag-3xFLAG by amplifying the human L30 promoter with prAH675 and prAH676 and assembling into AsiSI- (NEB R0630) and XbaI- (NEB R0145) digested PB PGKp-PuroR EF1a MCS-GDGAGLIN-HaloTag-3xFLAG. For the expression plasmid PB PGKp-PuroR EF1a 3x-FLAG-HaloTag-GDGAGLIN, we cloned three tandem copies of the SV40 nuclear localization sequence into XbaI- and BamHI-HF (NEB R3136)-digested PB PGKp-PuroR EF1a 3xFLAG-HaloTag-MCS using Gibson assembly.

For constructs expressing RARA-HaloTag domain deletions and point mutations, we first cloned the RARA coding sequence out of U2OS cDNA by extracting RNA from cycling U2OS cells with a QIAGEN RNeasy kit (QIAGEN 74104), preparing cDNA with the iScript Reverse Transcription Supermix (Bio-Rad 1708840), amplifying the CDS with prAH495 and prAH496, then assembling into an XbaI- and NotI-HF- (NEB R3189) digested PB PGKp-PuroR EF1a MCS-GDGAGLIN-HaloTag-3xFLAG using Gibson assembly. Next, to produce the mutants, we amplified parts of the RARA coding sequence in PCR fragments while introducing point mutations or domain deletions at the intersections of the fragments. PCR fragments were assembled into XbaI- and BamHI-HF-digested PB PGKp-PuroR L30p-MCS-GDGAGLIN-HaloTag-3xFLAG using Gibson assembly. The primers used for each construct were as follows: for PB PGKp-PuroR EF1a RARA[ΔNTD]-HaloTag-GDGAGLIN-3xFLAG, PCR fragment 1 was produced with prAH1111 and prAH1112; for PB PGKp-PuroR EF1a RARA[ΔCTD]-HaloTag-GDGAGLIN-3xFLAG, PCR fragment 1 was produced with prAH1113 and prAH1114; for PB PGKp-PuroR EF1a RARA[ΔNTD,ΔCTD]-HaloTag-GDGAGLIN-3xFLAG, PCR fragment 1 was produced with prAH1111 and prAH1114; for PB PGKp-PuroR EF1a RARA[C88G]-HaloTag-GDGAGLIN-3xFLAG, PCR fragment 1 was produced with prAH1113 and prAH1069 and PCR fragment 2 was produced with prAH1112 and prAH1070; for PB PGKp-PuroR EF1a RARA[ΔDBD]-HaloTag-GDGAGLIN-3xFLAG, PCR fragment 1 was produced with prAH596 and prAH704 and PCR fragment 2 was produced with prAH597 and prAH705; for PB PGKp-PuroR EF1a RARA[ΔLBD]-HaloTag-GDGAGLIN-3xFLAG, PCR fragment 1 was produced with prAH596 and prAH706 and PCR fragment 2 was produced with prAH597 and prAH707.

To generate the plasmid-based homology repair donor for gene editing at the human RARA exon 9 locus, we assembled the following fragments by Gibson assembly. For fragment 1, we digested the pUC57 vector with EcoRI and HindIII. For fragment 2, we amplified the left homology arm out of U2OS genomic DNA with prAH599 and prAH600. For fragment 3, we amplified the GDGAGLIN-HaloTag-3xFLAG insert out of the plasmid PB PGKp-PuroR L30p MCS-GDGAGLIN-HaloTag-3xFLAG with prAH601 and prAH602. For fragment 4, we amplified the right homology arm out of U2OS genomic DNA with prAH603 and prAH604.

To generate guide RNA/Cas9 expression plasmids for gene editing at the human RARA exon 9 locus, we cloned the two guide RNA sequences under a U6 promoter in a vector that coexpresses the sgRNA, mVenus, and *S. pyogenes* Cas9, which has been previously described (*Hansen et al., 2017*).

In luciferase assays, we used the retinoic acid-responsive firefly luciferase expression vector pGL3-RARE-luciferase (Addgene plasmid #13458; http://n2t.net/addgene:13458; RRID:Addgene_13458), a gift from T. Michael Underhill (*Hoffman et al., 2006*). Renilla luciferase was expressed from pRL CMV Renilla (Promega E2261).

## Cell lines

Human U2OS cells (female, 15 years old, osteosarcoma) obtained from the UC Berkeley Cell Culture Facility were cultured under 5% $CO_2$ at 37°C in DMEM containing 4.5 g/L glucose supplemented with 10% fetal bovine serum and 10 U/mL penicillin-streptomycin. Cells were subpassaged at a ratio of 1:6 every 3–4 days. The stable cell line expressing H2B-HaloTag-SNAPf was described previously (*Hansen et al., 2017*, *McSwiggen et al., 2019*). We induced exogenous expression of HaloTag, HaloTag-NLS, and point mutants and domain deletions of RARA-HaloTag by nucleofection of PiggyBac vectors containing the proteins under EF1a promoters. Expression of wildtype RARA-HaloTag and NPM1-HaloTag was induced by endogenous gene editing, as described in the 'CRISPR/Cas9-mediated gene editing' section. The U2OS cell line used here was validated by whole-genome sequencing as described in *Hansen et al., 2017*, and mycoplasma testing was performed by DAPI staining.

For sptPALM experiments, cells were grown on 25 mm circular No. 1.5 H coverglasses (Marienfeld, Germany, High-Precision 0117650) that were first sonicated in ethanol for 10 min, plasma-cleaned, then stored in isopropanol until use. U2OS cells were grown directly on the coverglasses in regular culture medium. The medium was changed after dye labeling and immediately before imaging into phenol red-free medium.

## Nucleofection

For all imaging experiments involving exogenous expression, we used the Lonza Amaxa II Nucleofector System with Cell Line Nucleofector Kit V reagent (Lonza VCA-1003). Briefly, U2OS cells were grown in 10 cm plates (Thermo Fisher) for 2 days prior to nucleofection, trypsinized, spun down at 1200 rpm for 5 min, combined with vector and Kit V reagent according to the manufacturer's instructions, and nucleofected with program X-001 on an Lonza Amaxa II Nucleofector. After nucleofection, cells were immediately resuspended in regular culture medium at 37°C and plated onto coverslips. In all imaging experiments involving nucleofection, imaging was performed within 24 hr of plating.

## CRISPR/Cas9-mediated gene editing

Endogenous tagging of RARA in U2OS cells was performed with a protocol roughly following *Hansen et al., 2017* with some modifications. A complete list of the plasmids used in gene editing is provided in *Supplementary file 2*, and a list of the guide sequences is provided in *Supplementary file 3*.

For U2OS cells, we nucleofected cells with plasmid expressing 3xFLAG-SV40NLS-pSpCas9 from a CBh promoter (*Ran et al., 2013*), mVenus from a PGK promoter, and guide RNA from a U6 promoter (pU6_sgRNA_CBh_Cas9_PGK_Venus_anti-RARA-C_terminus_1 and pU6_sgRNA_CBh_Cas9_PGK_Venus_anti-RARA-C_terminus_2), along with a second plasmid encoding the homology repair donor (pUC57_homRep_RARA-HaloTag). The homology repair donor was built in a pUC57 backbone modified to contain HaloTag-3xFLAG with ~500 base pairs of homologous genomic sequence on either side. Synonymous mutations were introduced at the cut site to prevent retargeting by Cas9. Each of the two guide RNA plasmids were nucleofected into separate populations of cells to be pooled for subsequent analysis. Then, 24 hr after the initial nucleofection, we screened for mVenus-expressing cells using FACS and pooled these mVenus-positive cells in 10 cm plates. Then, 5 days after plating, we labeled cells with HTL-TMR (Promega G8251) and screened for TMR-positive, mVenus-negative cells. Cells were diluted to single clones and plated in 96-well plates for a 2–3-week outgrowth step, during which the medium was replaced every 3 days. The 96-well plates were then screened for wells containing single colonies of U2OS cells, which were split by manual passage into two replicate wells in separate 96-well plates. One of these replicates was used to subpassage, while the other was used to harvest genomic DNA for PCR and sequencing-based screening for the correct homology repair product. In PCR-based genotyping, we used three primer sets: (A) primers external to the homology repair arms, expected to amplify both the wildtype allele and the edited allele ('PCR1'), (B) a primer internal to HaloTag and another external to it on the 5' side, expected to amplify only the edited allele ('PCR2'), and (C) a primer internal to HaloTag and another external to it on the 3' side, expected to amplify only the edited allele ('PCR3'). The primer sets for each target were the following. For RARA-GDGAGLIN-HaloTag-3xFLAG, we used prAH586 and prAH761 for PCR1, prAH761 and prAH762 for PCR2, and prAH763 and prAH764 for PCR3. For NPM1-GDGAGLIN-HaloTag-3xFLAG, we used prAH1092 and prAH1093 for PCR1, prAH1093 and prAH377 for PCR2, and prAH1092 and prAH373 for PCR3. U2OS cDNA from selected clones was isolated with DirectPCR Lysis Reagent (Viagen 101T),

treated with 0.5 mg/mL proteinase K for 15 min, incubated at 95°C for 1 hr, then subjected to PCRs 1–3 using Phusion polymerase in the presence of 5% DMSO. Amplicons from candidate clones were gel-purified (QIAGEN 28704) and Sanger sequenced; only clones with the correct target sequence were kept for continued screening. A subset of these clones were chosen for characterization by Western blot, imaging, and luciferase assays.

For NPM1-GDGAGLIN-HaloTag-3xFLAG knock-in cell lines, we used a different strategy relying on nucleofected *Streptococcus pyogenes* Cas9 sgRNPs and linear dsDNA homology repair donors. The target insert (GDGAGLIN-HaloTag-3xFLAG from the vector PB PGKp-PuroR L30p MCS-GDGAGLIN-HaloTag-3xFLAG) was first amplified with ultramers encoding 120 bp homology arms (prAH867 and prAH868; IDT) using KAPA2G Robust HotStart polymerase (Kapa Biosystems KR0379) for 12 cycles. A small volume of this reaction was then used to seed a PCR reaction using primers prAH869 and prAH870 in Q5 High-Fidelity 2X Master Mix (QIAGEN M0492). Products were purified by RNAClean XP magnetic beads (Beckman-Coulter A63987) and further cleaned by ethanol precipitation, followed by resuspension in a small volume of RNase-free water. For guides, we performed a three-primer PCR using prAH2000 and prAH2001 along with a unique oligo encoding the spacer (either prAH979 or prAH980) to produce a linear dsDNA product encoding the sgRNA preceded by a T7 promoter. We then used T7 RNA polymerase (NEB E2040S) to transcribe sgRNA from this template and purified the sgRNA with RNAClean XP magnetic beads according to the manufacturer's instructions. To assemble the sgRNP, we incubated 80 pmol sgRNA with 40 pmol purified SpyCas9-NLS (UC Berkeley Macrolab) for 15 min at 37°C in 20 mM HEPES pH 7.5, 150 mM KCl, 10 mM $MgCl_2$, and 5% glycerol. sgRNPs were subsequent kept on ice and combined with donor immediately before nucleofection. For each nucleofection, we used 40 pmol sgRNP and 5 pmol dsDNA donor template suspended in <10 μL with Lonza Amaxa Nucleofector II protocol X-001 in Lonza Kit V reagent. Roughly 1 million cells were used for nucleofection. Sorting for labeled cells, subcloning, and genotyping proceeded as previously described for RARA-GDGAGLIN-HaloTag-3xFLAG.

## Western blots

Antibodies were as follows (the ratio indicates the dilution factors used for Western blot): human TBP, Abcam Ab51841, 1:2500 (mouse); FLAG, Sigma-Aldrich F3165, 1:2000 (mouse).

For Western blots, cells were scraped from plates in ice-cold PBS then pelleted. Pellets were resuspended in lysis buffer (0.15 M NaCl, 1% NP-40, 50 mM Tris–HCl [pH 8.0], and a cocktail of protease inhibitors [Sigma-Aldrich 11697498001 dissolved in PBS with supplemented PMSF, aprotinin, and benzamidine]), agitated for 30 min at 4°C, then centrifuged for 20 min at 12,000 rpm, 4°C. The supernatant was then mixed with 2× Laemmli (to final 1×), boiled for 5 min, then run on 12.5% SDS-PAGE. After transfer to nitrocellulose, the membrane was blocked with 10% condensed milk in TBST (500 mM NaCl, 10 mM Tris–HCl [pH 7.4], 0.1% Tween-20) for 1 hr at room temperature. Antibodies were suspended in 5% condensed milk in TBST at the dilutions indicated above and incubated rocking at 4°C overnight. After primary hybridization, the membrane was washed three times for 10 min with TBST at room temperature, hybridized with an anti-mouse HRP secondary antibody in 5% condensed milk in TBST for 60 min at room temperature, washed three more times with TBST for 10 min, then visualized with Western Lightning Plus-ECL reagent (PerkinElmer NEL103001) according to the manufacturer's instructions and imaged on a Bio-Rad ChemiDoc imaging system. Different exposure times were used for each antibody. The raw Western blots images for RARA-HaloTag and NPM1-HaloTag are provided as *Figure 5—source data 1* and *Figure 6—figure supplement 1—source data 1* , respectively.

## Luciferase assays

All luciferase assays used pGL3-RARE-luciferase, a reporter containing firefly luciferase driven by an SV40 promoter with three retinoic acid response elements (RAREs). pGL3-RARE-luciferase was a gift from T. Michael Underhill (Addgene plasmid 13458; http://n2t.net/addgene:13458; RRID:Addgene_13458; *Hoffman et al., 2006*). Luciferase assays were performed on cells cultivated in 6-well plates. Cells were transfected with 100 ng pGL3-RARE-luciferase and 10 ng pRL Renilla (Promega E2261) using Mirus TransIT-2020 Transfection Reagent (Mirus MIR 5404). Transfection was performed 1 day before assaying luciferase expression with the Dual-Luciferase Reporter Assay System (Promega

E1910) according to http://n2t.net/addgene:13458 the manufacturer's instructions. Readout was performed on a GloMax luminometer (Promega).

## Cell dye labeling

For sptPALM experiments, cells were labeled with one of two methods, depending on the dye. For non-photoactivatable fluorescent dyes including TMR-HTL (tetramethylrhodamine-HaloTag ligand; Promega G8251), we stained cells with 100 nM dye in regular culture medium for 10 min, then performed three 10 min incubations in dye-free culture medium separated by PBS washes. All PBS and culture medium was incubated at 37°C between medium changes and washes.

For experiments with photoactivatable dyes, which have lower cell permeability and slower wash-in/wash-out kinetics, we labeled cells with 100 nM dye in regular culture medium for 10–20 min, followed by four 30 min incubations in dye-free culture medium at 37°C. Between each incubation, we washed twice with PBS at 37°C. After the final incubation, cells were changed into phenol red-free medium for imaging.

## sptPALM

sptPALM experiments were performed with a custom-built Nikon TI microscope equipped with a ×100/NA 1.49 oil-immersion TIRF objective (Nikon apochromat CFI Apo TIRF 100X Oil), an EMCCD camera (Andor iXon Ultra 897), a perfect focus system to account for axial drift, an incubation chamber maintaining a humidified 37°C atmosphere with 5% $CO_2$, and a laser launch with 405 nm (140 mW, OBIS, Coherent), 488 nm, 561 nm, and 633 nm (all 1 W, Genesis Coherent) laser lines. Laser intensities were controlled by an acousto-optic Tunable Filter (AA Opto-Electronic, AOTFnC-VIS-TN) and triggered with the camera TTL exposure output signal. Lasers were directed to the microscope by an optical fiber, reflected using a multi-band dichroic (405 nm/488 nm/561 nm/633 nm quad-band, Semrock) and focused in the back focal plane of the objective. The angle of incident laser was adjusted for highly inclined laminated optical sheet (HiLo) conditions (*Tokunaga et al., 2008*). Emission light was filtered using single band-pass filters (Semrock 593/40 nm for PAJFX549 and Semrock 676/37 nm for PAJF646). Hardware was controlled with the Nikon NIS-Elements software.

For stroboscopic illumination, the excitation laser (561 nm or 633 nm) was pulsed for 1–2 ms (most commonly 1 ms) at maximum (1 W) power at the beginning of the frame interval, while the photoactivation laser (405 nm) was pulsed during the ~447 μs camera transition time, so that the background contribution from the photoactivation laser is not integrated. For all sptPALM, we used an EMCCD vertical shift speed of 0.9 μs and conversion gain setting 2. On our setup, the pixel size after magnification is 160 nm and the photon-to-grayscale gain is 109. A total of 15,000–30,000 frames with this sequence were collected per nucleus, during which the 405 nm intensity was manually tuned to maintain low density of fluorescent particles per frame.

## Localization and tracking

To produce trajectories from raw sptPALM movies, we used a custom sptPALM tracking pipeline publicly available on GitHub (https://github.com/alecheckert/quot, swh:1:rev:1adf7a0574c62f38140f-1dec2d14555bfc03b9a7, *Heckert, 2022b*). All localization and tracking for this article was performed with the following settings:

- Detection: Generalized log likelihood ratio test (*Sergé et al., 2008*) with a 2D Gaussian kernel with $\sigma = 190$ nm (detection method "llr" with k = 1.2, a 15 pixel window size [w = 15], and a log ratio threshold of 16.0 [t = 16.0]).

- Subpixel localization: Levenberg–Marquardt fitting of a 2D integrated Gaussian point spread function model (localization method "ls_int_gaussian") with fixed $\sigma = 190$ nm, window size 9 pixels, maximum 20 iterations per PSF, with a damping term of 0.3 for parameter updates. The 2D integrated Gaussian PSF model is described in *Smith et al., 2010* and the Levenberg–Marquardt routine in *Laurence and Chromy, 2010*. We used the radial symmetry method (*Parthasarathy, 2012*) to make the initial guess used to start the Levenberg–Marquardt algorithm.

- Tracking: We used the tracking algorithm "conservative" in quot with a 1.2 μm search radius. This simple algorithm searches for particle–particle reconnections that are 'unambiguous' in the sense that no other reconnections are possible within the specified search radius. These

reconnections are then used to synthesize trajectories, while 'ambiguous' connections are discarded.

After localization and tracking, all trajectories in the first 1000 frames of each movie were discarded. Localization density tends to be high in these frames, so they can contribute tracking errors that compromise accuracy. The mean localization density for most movies in the remaining set of frames was less than one emitter per frame.

For experiments involving HaloTag or HaloTag-NLS, which have high mobility, we used a broader search radius at 2.5 μm. All other settings were kept the same.

All trajectories from real sptPALM experiments used in this article are publicly accessible as a Dryad dataset (https://doi.org/10.6078/D13H6N).

## Spinning disk confocal imaging

Experiments using spinning disk confocal imaging were performed at the UC Berkeley High-Throughput Screening Facility on a Perkin Elmer Opera Phenix equipped with a controller for 37°C and 5% $CO_2$, using a built-in 40x water immersion objective.

## Simulations

All simulations in this article belong to one of two categories:

- Trajectory simulations: Individual trajectories are simulated in a HiLo-like geometry, then localization error and defocalization are injected as detailed below. The output of each simulation is a set of trajectories. This type of simulation does not incorporate tracking errors. These include the simulations in *Figure 3*, *Figure 4*, *Figure 4—figure supplement 1*, *Figure 4—figure supplement 2*, *Figure 4—figure supplement 3*, *Figure 4—figure supplement 4*, *Figure 4—figure supplement 7*, *Figure 4—figure supplement 8*, and *Figure 4—figure supplement 9*.
- Optical–dynamical simulations: Starting from a dynamical model and an approximation to sptPALM imaging system, we simulate full SPT movies. The output is a stack of images similar to that acquired on a real sptPALM system. Analysis follows the same steps as for real SPT data: we recover trajectories using a localization and tracking algorithm, which are subjected to the relevant downstream analyses. These include the simulations in *Figure 1—figure supplement 1*, *Figure 4—figure supplement 5*, *Figure 4—figure supplement 6*, *Figure 4—figure supplement 10*, *Figure 4—figure supplement 11*, and *Figure 4—figure supplement 12*.

Both types of simulation are important. Trajectory simulations allow us to separate the accuracy of the tracking algorithm from the accuracy of the SA/DPMM algorithm in a tightly controlled setting, while optical–dynamical simulations are 'end-to-end' tests that also incorporate realistic features such as motion blur, camera noise, and tracking errors.

### Trajectory simulations

All trajectory simulations were performed with a simple publicly available sptPALM simulation tool (`strobesim`; https://github.com/alecheckert/strobesim, *Heckert, 2022e*). This tool generates trajectories for different types of motion and simulates the act of observation in a thin focal plane.

Unless otherwise noted, simulated trajectories were confined to a sphere with radius 5 μm and a focal plane with 700 nm depth bisecting the sphere. Simulated particles were subject to photoactivation and photobleaching throughout the sphere and were only observed when their positions coincided with the focal volume. We simulated sparse tracking without gaps, so that if an emitter passed twice through the focal volume, it counted as two separate trajectories. At the sparsity used for these simulations, tracking is unambiguous and so tracking errors do not contribute to the outcome.

For discrete-state trajectory simulations, the number of particles in each state was modeled as a multinomial random variable drawn from the underlying state occupancies. As a result, there is an inherent variability associated with the 'true' fractional occupations for each simulation replicate, exactly as would be expected in sptPALM experiments.

For trajectory simulations with state transitions, we modeled the particles as two-state Markov chains with identical transition rates between the states. Each state was associated with a constant diffusion coefficient. These Markov chains were simulated on subframes grained at 100 iterations per frame interval. For instance, for simulations with 7.48 ms frame intervals, the underlying Markov chain was simulated on subframes of 74.8 μs. During each subframe, the state of the MC was assumed to

be constant and we simulated diffusion according to the Euler–Maruyama scheme with the current diffusion coefficient. The positions of the particle at the frame interval were recorded.

## Optical–dynamical simulations

The simulations in *Figure 1—figure supplement 1*, *Figure 4—figure supplement 5*, *Figure 4—figure supplement 6*, *Figure 4—figure supplement 10*, *Figure 4—figure supplement 11*, and *Figure 4—figure supplement 12* were produced with a software package (`sptPALMsim`, https://github.com/alecheckert/sptpalmsim, *Heckert, 2022d*) that performs both dynamical and optical simulations to incorporate effects such as defocus, camera noise, motion blur, and tracking errors. The dynamic simulations are identical to those described in the previous section. Here, we outline the optical simulations. A more detailed discussion can be found, for instance, in *Hanser et al., 2004*.

We assume that the observed intensity $I_{ij}$ on pixel $i,j$ is produced by a linear gain model with read noise and shot noise:

$$
\begin{aligned}
I_{ij} &= b + g n_{ij} + r_{ij} \\
n_{ij} &\sim \text{Poisson}\left(A_{ij}\right) \\
r_{ij} &\sim \mathcal{N}\left(0, \sigma_{\text{read noise}}^2\right)
\end{aligned}
\tag{1}
$$

The offset $b$, gain $g$, and read noise variance $\sigma_{\text{read noise}}^2$ are assumed to be the same for all pixels in the camera with values similar to an Andor iXon 897 EMCCD ($b = 470$, $g = 109$, and $\sigma_{\text{read noise}}^2 = 3^2$). The function $A_{ij}$ defines the rate of photon arrivals at pixel $i,j$ and depends on the distribution of fluorescent emitters in the sample.

We assume that the photon arrival rate $A_{ij}$ is related to the distribution of emitters in the source plane $f(x, y, z)$ via

$$
A_{ij} = \iint\limits_{\text{pixel } i,j} \mathrm{d}x\,\mathrm{d}y \int\limits_{-\Delta z/2}^{+\Delta z/2} \mathrm{d}z\, f(x, y, z) * \text{PSF}(x, y, z)
\tag{2}
$$

* denotes convolution. The $z$-integral runs over the depth of the simulation (in this article, this is always from $z = -2$ μm to $z = +2$ μm). $\text{PSF}(x, y, z)$ is assumed to be given by the squared magnitude of a complex-valued function $\text{PSF}_A(x, y, z)$ such that (*Hanser et al., 2004*):

$$
\begin{aligned}
\text{PSF}_A(x, y, z) &= \iint\limits_{\text{pupil}} P(k_x, k_y) e^{i(k_x x + k_y y)} e^{i k_z(k_x, k_y) z}\, \mathrm{d}k_x \mathrm{d}k_y \\
k_z(k_x, k_y) &= \sqrt{\left(\frac{2\pi n}{\lambda}\right)^2 - \left(k_x^2 + k_y^2\right)}
\end{aligned}
\tag{3}
$$

where $P(k_x, k_y)$ is the complex-valued microscope pupil function and $e^{i k_z(k_x, k_y) z}$ is a 'defocus kernel,' accounting for the phase profile of light exiting the pupil plane. The limits of the integral run over the circular microscope aperture $k_y^2 + k_x^2 \leq \left(\frac{\text{NA}}{\lambda}\right)^2$, where $\lambda$ is the emission wavelength. In all simulations, we use an 'ideal' pupil function with phase 0 and amplitude 1 over the microscope aperture.

For our purposes, the integral in 3 is replaced with a sum over a grid with finer spatial grain than the camera pixel size.

Altogether, the optical simulations proceeded in the following way:

1. First, the paths of fluorescent emitters are simulated with fine temporal grain (such as $10^4$ Hz) according to a particular dynamic model.
2. For each laser pulse:
   a. The parts of the emitter paths that temporally coincide with the laser pulse are aggregated into a single distribution $f(x, y, z)$.
   b. The photon arrival rates at the camera is simulated according to *Equation 2*.
   c. Shot noise and read noise are introduced according to *Equation 1*.
3. Images for all laser pulses are concatenated to yield the simulated SPT movie.

The product of these simulations are SPT movies that are subsequently tracked (see 'Localization and tracking'). Except where otherwise indicated, the settings for these simulations were as follows: numerical aperture 1.49, immersion media refractive index 1.515, emission wavelength 670 nm, frame

interval 7.48 ms, image pixel size 0.16 µm, excitation pulse width 2 ms, bleach rate 0.2 Hz, read noise variance $3^2$ grayvalues$^2$, offset 470.0 grayvalues, and gain 109.0 grayvalues per photon. The mean number of photons detected per emitter per frame was 150, although the actual number is random due to the randomness of photon emission and detection (*Equation 1*). The scripts used to generate the simulations are publicly available at the `sptPALMsim` repo (https://github.com/alecheckert/sptpalmsim). *Video 4* shows an example of a movie simulated with these settings.

## State arrays and Dirichlet process mixture models

This section describes the SA and DPMM used in this article. We begin with a classic Bayesian finite state mixture model, then introduce modifications that lead to SAs and DPMMs. The finite state mixture has been reviewed in detail elsewhere (*Marin et al., 2005*, *McLachlan et al., 2019*), so here we keep details to a minimum.

### Finite state mixtures

A finite state mixture is a collection of 'states' $k = 1, ..., K$, each of which is associated with an occupation $\tau_k$ and a vector of state parameters $\boldsymbol{\theta}_k$. (Where convenient, we let $\boldsymbol{\Theta} = (\boldsymbol{\theta}_1, ..., \boldsymbol{\theta}_K)$ be the collection of parameters for all states.) Each state generates trajectories $X$ according to some distribution $p_X(x|\boldsymbol{\theta}_k)$. The overall generative process for each trajectory is:

1. Randomly select a state $k$ with probability $\tau_k$.
2. Randomly generate a trajectory $X$ from that state according to $p_X(x|\boldsymbol{\theta}_k)$.

The probability to generate a particular trajectory $X$ is then

$$p(X|\boldsymbol{\tau}, \boldsymbol{\theta}_1, ..., \boldsymbol{\theta}_K) = \sum_{k=1}^{K} \tau_k p_X(x|\boldsymbol{\theta}_k)$$

To represent the origin state for each trajectory, we use a 1-of-$K$ encoding $\boldsymbol{Z}_i \in \{0,1\}^K$ so that $Z_{ik} = 1$ if trajectory $i$ originates from state $k$ and $Z_{ik} = 0$ otherwise. For a dataset with $N$ trajectories, we let $\boldsymbol{Z} \in \{0,1\}^{N \times K}$ be the matrix such that the $i$th row is $\boldsymbol{Z}_i$.

Finally, we specify priors over $\boldsymbol{\tau}$ and $\boldsymbol{\theta}_k$. The full Bayesian finite state mixture can then be written as

$$
\begin{aligned}
\boldsymbol{\tau} &\sim \text{Dirichlet}\left(\tfrac{\alpha}{K}, ..., \tfrac{\alpha}{K}\right) \\
\boldsymbol{\theta}_k &\sim H \\
\boldsymbol{Z}_i|\boldsymbol{\tau} &\sim \text{Multinomial}\left(\boldsymbol{\tau}, 1\right) \\
X_i|Z_{ik} = 1, \boldsymbol{\theta}_k &\sim p_X(x|\boldsymbol{\theta}_k)
\end{aligned}
\tag{4}
$$

where $H$ is the prior over the parameters $\boldsymbol{\theta}_k$, usually chosen to be conjugate to $p_X(x|\boldsymbol{\theta}_k)$.

This corresponds to the first graphical model in *Figure 2—figure supplement 1*. The objective is to infer the posterior distribution $p(\boldsymbol{Z}, \boldsymbol{\tau}, \boldsymbol{\Theta}|X)$, where $X$ represents some observed set of trajectories.

### State arrays

Three common challenges with the finite state mixture 4 are:

1. Choosing $K$, the number of states. Because $K$ is a hyperparameter in 4, some kind of meta-algorithm is required to infer it, and this process can be fraught (*Marin et al., 2005*).
2. Choosing $H$, the prior over $\boldsymbol{\theta}_k$. Ideally the prior is chosen to be conjugate to $p_X(x|\boldsymbol{\theta}_k)$, but this is only possible for the simplest forms of $p_X(x|\boldsymbol{\theta}_k)$.
3. Computing $p_X(x|\boldsymbol{\theta}_k)$ is often expensive, especially if it needs to be evaluated repeatedly during inference.

SAs are a special case of finite mixture models designed in response to these issues. Rather than equating $K$ with the true number of states, SAs instead choose a large, fixed value of $K$ and constant values for each $\boldsymbol{\theta}_k$. A Bayesian routine is then used to drive the occupation of most states to zero, leaving minimal models sufficient to explain the observations. (The ability of Bayesian inference to identify sparse explanatory models in the presence of more complex alternatives is the same property that drives automatic relevance determination [ARD] in machine learning with Bayesian models.)

Because the state parameters are constant, the only parameters left to infer are $\boldsymbol{Z}$ and $\boldsymbol{\tau}$. Together, this simplified model is

$$
\begin{aligned}
\boldsymbol{\tau} &\sim \text{Dirichlet}\left(\tfrac{\alpha}{K}, ..., \tfrac{\alpha}{K}\right) \\
\boldsymbol{Z}_i \mid \boldsymbol{\tau} &\sim \text{Multinomial}(\boldsymbol{\tau}, 1) \\
X_i \mid Z_{ik} = 1 &\sim p_X(x \mid \boldsymbol{\theta}_k)
\end{aligned}
\tag{5}
$$

This corresponds to the third graphical model shown in *Figure 2A*. Notice that since each $X_i$ and $\boldsymbol{\theta}_k$ are constant, $p_X(X_i \mid \boldsymbol{\theta}_k)$ is also constant and only needs to be evaluated once during inference.

To infer the posterior distribution $p(\boldsymbol{Z}, \boldsymbol{\tau} \mid \boldsymbol{X})$, we take a variational approach, constructing an approximation $q(\boldsymbol{Z}, \boldsymbol{\tau}) \approx p(\boldsymbol{Z}, \boldsymbol{\tau} \mid \boldsymbol{X})$ such that

$$
\begin{aligned}
q(\boldsymbol{Z}, \boldsymbol{\tau}) &= q(\boldsymbol{Z})q(\boldsymbol{\tau}) \\
q(\boldsymbol{Z}, \boldsymbol{\tau}) &= \underset{\boldsymbol{Z}, \boldsymbol{\tau}}{\text{argmax}}\, L[q]
\end{aligned}
\tag{6}
$$

where $L[q]$ is the variational lower bound:

$$
L[q] = \sum_{\boldsymbol{Z}} \int q(\boldsymbol{Z}, \boldsymbol{\tau}) \log\left[\frac{p(\boldsymbol{X}, \boldsymbol{Z}, \boldsymbol{\tau})}{q(\boldsymbol{Z}, \boldsymbol{\tau})}\right] \mathrm{d}\boldsymbol{\tau}
\tag{7}
$$

Motivation for the variational lower bound is discussed in detail elsewhere (*Bishop, 2006*). Here, we only remark that maximization of $L[q]$ minimizes the Kullback–Leibler divergence between the approximation and the true posterior. The factorability criterion in 6 enables an expectation-maximization routine (*Dempster et al., 1977*) by iteratively evaluating.

$$
\begin{aligned}
\log q(\boldsymbol{Z}) &= E_{\boldsymbol{\tau} \sim q(\boldsymbol{\tau})}\left[\log p(\boldsymbol{X}, \boldsymbol{Z}, \boldsymbol{\tau})\right] + \text{constant} \\
\log q(\boldsymbol{\tau}) &= E_{\boldsymbol{Z} \sim q(\boldsymbol{Z})}\left[\log p(\boldsymbol{X}, \boldsymbol{Z}, \boldsymbol{\tau})\right] + \text{constant}
\end{aligned}
\tag{8}
$$

The constants are chosen so that each factor, $q(\boldsymbol{Z})$ or $q(\boldsymbol{\tau})$, is normalized. Combining *Equations 8* for model 5 yields the solution

$$
\begin{aligned}
q(\boldsymbol{Z}) &= \prod_{i=1}^{N} \prod_{k=1}^{K} r_{ik}^{Z_{ik}} \\
q(\boldsymbol{\tau}) &= \text{Dirichlet}\left(n_1, ..., n_K\right) \\
r_{ik} &= \frac{A_{ik} e^{\psi(n_k)}}{\sum_{j=1}^{K} A_{ij} e^{\psi(n_j)}} \\
n_k &= \frac{\alpha}{K} + \sum_{k=1}^{K} L_i r_{ik}
\end{aligned}
\tag{9}
$$

where $L_i$ is the number of jumps in trajectory $i$ and $\psi(n)$ is the digamma function. For brevity here, the derivation of *Equation 9* is placed in its own section below.

$q(\boldsymbol{Z}, \boldsymbol{\tau})$ is parameterized by $\boldsymbol{n}$ and $\boldsymbol{r}$. These can be inferred with a simple EM algorithm:

1. Initialize a matrix $A \in R^{N \times K}$ by setting $A_{ik} = p(X_i \mid \boldsymbol{\theta}_k)$.
2. Initialize a matrix $\boldsymbol{r}^{(0)} \in R^{N \times K}$ such that $r_{ik}^{(0)} = \frac{A_{ik}}{\sum_{j=1}^{K} A_{ij}}$.

3. For each iteration $t = 1, 2, ...$:
   a. For each state $k$, evaluate $n_k^{(t)} = \frac{\alpha}{K} + \sum_{i=1}^{N} L_i r_{ik}^{(t-1)}$.
   b. Evaluate the matrix $\boldsymbol{r}^{(t)}$ such that
   
   $$
   r_{ik}^{(t)} = \frac{A_{ik} e^{\psi(n_k^{(t)})}}{\sum_{j=1}^{K} A_{ij} e^{\psi(n_j^{(t)})}}
   $$

4. After convergence of $\boldsymbol{n}$ and $\boldsymbol{r}$, we can summarize the posterior distribution by taking its mean:

$$\begin{aligned} E\left[Z_{ik}\right] &= r_{ik} \\ E\left[\tau_k\right] &= \frac{n_k}{\sum_{j=1}^{K} n_j} \end{aligned} \tag{10}$$

5. Finally, we perform two postprocessing steps on the posterior mean:
   a. If localization error is a parameter, we marginalize it out by projecting through that axis of the array.
   b. We adjust the posterior mean to account for defocalization biases, as described in 'Defocalization'.

Throughout this article, we always report occupations for the SA model as the mean of $q(\boldsymbol{\tau})$ according to *Equation 10*, twith localization error marginalized out and the appropriate defocalization correction applied.

## Naive state occupations

Inference of the SA posterior works optimally with thousands to tens of thousands of trajectories. We also found it useful to have a cheap, dirty estimate for state occupations that can be evaluated on a small number of trajectories to visualize nuclei-to-nuclei variability (for instance, in *Figure 5A*).

For these purposes, we define the 'naive occupation estimate' $\tau_{\text{naive}}$ such that

$$\begin{aligned} \tau_{\text{naive},k} &= \frac{\sum_{i=1}^{N} L_i r_{ik}^{(0)}}{\sum_{j=1}^{K} \sum_{i=1}^{N} L_i r_{ij}^{(0)}} \\ r_{ik}^{(0)} &= \frac{p_X(X_i|\boldsymbol{\theta}_k)}{\sum_{j=1}^{K} p_X(X_i|\boldsymbol{\theta}_j)} \end{aligned} \tag{11}$$

Notice that this is just the posterior occupations based on the initial value for $\boldsymbol{r}$ in the algorithm for SA inference. We use the same postprocessing steps for $\tau_{\text{naive}}$ as for SAs, including marginalizing out localization error and correcting for defocalization.

## State arrays for regular Brownian motion

In the above section, we have left $p_X(x|\boldsymbol{\theta}_k)$ unspecified as it depends on the type of motion being considered. This section states the form of $p_X(x|\boldsymbol{\theta}_k)$ for RBME, the type of motion considered in this article.

Suppose that trajectory $i$ is constructed by measuring the position of a Brownian particle over sequential frame intervals of duration $\Delta t$, and that each measured position has some error associated with it. We assume that this error is normally distributed with mean zero and variance $\sigma_{\text{loc}}^2$.

We refer to the change in the particle's position over each frame interval as a 'jump.' If there are $L_i$ total jumps, let $\boldsymbol{x}, \boldsymbol{y} \in R^{L_i}$ be the displacements of these jumps along the $x$ and $y$ axes, respectively. Then, the probability density over $\boldsymbol{x}$ and $\boldsymbol{y}$ is

$$p_X(\boldsymbol{x}, \boldsymbol{y}|D, \sigma_{\text{loc}}^2) = \frac{\exp\left(-\frac{1}{2}\left[\boldsymbol{x}^T \boldsymbol{\Gamma}^{-1} \boldsymbol{x} + \boldsymbol{y}^T \boldsymbol{\Gamma}^{-1} \boldsymbol{y}\right]\right)}{2\pi \det(\boldsymbol{\Gamma})} \tag{12}$$

where $\boldsymbol{\Gamma} \in R^{L_i \times L_i}$ is the covariance matrix defined by

$$\Gamma_{ij} = \begin{cases} 2(D\Delta t + \sigma_{\text{loc}}^2) & \text{if } i = j \\ -\sigma_{\text{loc}}^2 & \text{if } |i - j| = 1 \\ 0 & \text{otherwise} \end{cases}$$

where $D$ is the diffusion coefficient and $\sigma_{\text{loc}}^2$ is the localization error (*Michalet and Berglund, 2012*). Due to the contribution of the localization error to the off-diagonal terms of the covariance matrix, the jumps of an RBME are not a Markov process except when $\sigma_{\text{loc}}^2 = 0$.

The SA for RBME uses a 2D grid of diffusion coefficients and localization errors. In this grid, the diffusion coefficients $D$ are log-spaced between $10^{-2}$ and $10^2$ µm² s⁻¹, while the localization errors $\sigma_{\text{loc}}$ are linearly spaced between 0 and 0.06 µm.

## State arrays for fractional Brownian motion

In *Figure 4—figure supplement 11* and *Figure 4—figure supplement 12*, we consider a generalization of RBME that we refer to as fractional Brownian motion with localization error (FBME). This is a simple modification of Mandelbrot and Van Ness's FBM (*Mandelbrot and Van Ness, 1968*) that incorporates localization error.

We define 1D FBME as a mean-zero Gaussian process $X_t$ with the covariance function

$$\text{Cov}(X_t, X_s) = S\left(|t|^{2H} + |s|^{2H} - |t-s|^{2H}\right) + I_{t=s}\sigma_{\text{loc}}^2$$

where $S$ is the scaling coefficient, $H$ is the Hurst parameter ($0 < H < 1$), $\sigma_{\text{loc}}^2$ is the variance of the localization error, and $I_{t=s}$ is the indicator function (1 if $t = s$ and 0 otherwise). Because we always measure the position at regular frame intervals of duration $\Delta t$, we let $t = i\Delta t$ and $s = j\Delta t$ so that this can be written as

$$\text{Cov}(X_{i\Delta t}, X_{j\Delta t}) = S\Delta t^{2H}\left(|i|^{2H} + |j|^{2H} - |i-j|^{2H}\right) + I_{i=j}\sigma_{\text{loc}}^2$$

The corresponding increment process $\tilde{X}_i = X_{i\Delta t} - X_{(i-1)\Delta t}$ is a mean-zero Gaussian process with the covariance function

$$\begin{aligned}
\text{Cov}(\tilde{X}_i, \tilde{X}_j) \quad &= S\Delta t^{2H}\left(|i-j+1|^{2H} + |i-j-1|^{2H} - 2|i-j|^{2H}\right) \\
&\quad + (2\sigma_{\text{loc}}^2)I_{i=j} - \sigma_{\text{loc}}^2 I_{|i-j|=1}
\end{aligned} \tag{13}$$

2D and 3D FBMEs are constructed with independent 1D FBMEs along each spatial axis.

In *Equation 13*, the scaling coefficient has units of $\mu\text{m}^2\text{s}^{-2H}$. As a result, its magnitude is highly dependent on $H$. Because we often want to parameterize the magnitude of the particle's jumps separately from the covariance between jumps, in this article we use a 'modified' scaling parameter $\bar{S}$ defined by

$$\bar{S} = S\Delta t^{2H-1} \tag{14}$$

As a result, the jump variance is $\text{Var}(\tilde{X}_i) = 2\bar{S}\Delta t$, regardless of the Hurst parameter. While $\bar{S}$ is much easier to work with for one dataset, since it is dependent on $\Delta t$ it must not be compared across datasets with different frame intervals and should first be converted to $S$ with *Equation 14*.

## Derivation of Equation 9

Here, we derive the SA posterior (*Equation 9*) by substituting model 5 into *Equation 8* and imposing some additional physical constraints.

First, let $A_{ik} = p_X(X_i|\theta_k)$. Then factor $\log p(X, Z, \tau)$ as

$$\begin{aligned}
\log p(X, Z, \tau) \quad &= \log p(X|Z) + \log p(Z|\tau) + \log p(\tau) \\
&= \sum_{k=1}^{K}\sum_{i=1}^{N} Z_{ik}\log A_{ik} + \sum_{k=1}^{K}\sum_{i=1}^{N} Z_{ik}\log\tau_k + \sum_{k=1}^{K}(\alpha - 1)\log\tau_k \\
&\quad + \text{constant}
\end{aligned} \tag{15}$$

where the constant accounts for normalization factors. Plugging this into the second equation in Equation 8, we have

$$\log q(\tau) = \sum_{k=1}^{K}\left(\frac{\alpha}{K} - 1 + \sum_{i=1}^{N} E_{Z \sim q(Z)}\left[Z_{ik}\right]\right)\log\tau_k + \text{constant}$$

We have collected terms that do not depend on $\tau$ into the constant. In this article, we choose to weight the contribution of each trajectory to $\log q(\tau)$ by the number of jumps in the trajectory. This is equivalent to treating jumps (rather than trajectories) as individual observations and is more robust to issues arising from the shallow observation depth of most sptPALM setups. It results in the modified equation

$$\log q(\boldsymbol{\tau}) = \sum_{k=1}^{K} \left( \frac{\alpha}{K} - 1 + \sum_{i=1}^{N} L_i E_{\mathbf{Z} \sim q(\mathbf{Z})} \left[ Z_{ik} \right] \right) \log \tau_k + \text{constant}$$

where $L_i$ is the number of jumps in trajectory $i$. We recognize this as a log Dirichlet distribution, so that

$$\begin{aligned} q(\boldsymbol{\tau}) &= \text{Dirichlet}\left(n_1, ..., n_K\right) \\ n_k &= \frac{\alpha}{K} + \sum_{i=1}^{N} L_i E\left[Z_{ik}\right] \end{aligned} \tag{16}$$

Next, we substitute *Equation 15* into the first equation in Equation 8, giving

$$\log q(\mathbf{Z}) = \sum_{k=1}^{K} \sum_{i=1}^{N} \left( \log A_{ik} + E_{\boldsymbol{\tau} \sim q(\boldsymbol{\tau})} \left[ \log \tau_k \right] \right) Z_{ik}$$

Since $q(\boldsymbol{\tau})$ is the Dirichlet distribution given by Equation 16,

$$E_{\boldsymbol{\tau} \sim q(\boldsymbol{\tau})} \left[ \log \tau_k \right] = \psi\left(n_k\right) - \psi\left(\sum_{j=1}^{K} n_j\right)$$

where $\psi(x)$ is the digamma function. Normalizing over the states for each trajectory $i$, we have

$$\begin{aligned} q(\mathbf{Z}) &= \prod_{i=1}^{N} \prod_{k=1}^{K} r_{ik}^{Z_{ik}} \\ r_{ik} &= \frac{A_{ik} e^{\psi(n_k)}}{\sum_{j=1}^{K} A_{ij} e^{\psi(n_j)}} \end{aligned} \tag{17}$$

Together, *Equations 16 and 17* constitute the result in Equation 9.

## Dirichlet process mixture model

As mentioned above, a fundamental challenge with the finite state mixture (*Equation 4*) is determining the number of states. SAs deal with this issue by selecting a large, finite value for $K$ and relying on an inference routine that selects sparse subsets of states from a $K$-dimensional initial model.

DPMMs are more extreme, taking the limit $K \to \infty$ (*Ferguson, 1973*). In this limit, the discrete vector of state occupations is replaced by a continuous distribution over the entire space of state parameters. The generative process for each trajectory is,

1. Randomly draw some state parameters $\boldsymbol{\theta}_i \sim H$, where $H$ is a continuous distribution over the space of state parameters.
2. Randomly generate a trajectory $X$ from that state according to $p_X(x|\boldsymbol{\theta}_i)$.

This process is formalized by replacing the Dirichlet distribution in Equation 4 with the Dirichlet process $\text{DP}(\alpha, H)$, its infinite-dimensional analog. Here, α has the same function as in the finite mixture (defining the relative strength of the prior) and $H$ is the 'base distribution' over state parameters. The full DPMM is then

$$\begin{aligned} G &\sim \text{DP}\left(\alpha, H\right) \\ \boldsymbol{\theta}_i \mid G &\sim G \\ X_i &\sim p_X(x|\boldsymbol{\theta}_i) \end{aligned} \tag{18}$$

This corresponds to the second graphical model in *Figure 2A*. Each draw $G$ is a discrete probability distribution over part of the parameter space (*Blackwell, 1973*). This formalism is discussed in detail in *Teh, 2010* or *Neal, 1992*. Here, we only remark that recovering the posterior $p(\boldsymbol{\theta}|X)$ requires marginalizing over $G$, yielding a continuous distribution over the parameter space.

To estimate the posterior distribution $p(\boldsymbol{\theta}|X)$, we take the Gibbs sampling approach introduced by Neal (Algorithm 8 in *Neal, 2000*). This involves sampling each $\boldsymbol{\theta}_i$ while hold the other $\boldsymbol{\theta}_{j \neq i}$ constant, yielding samples from the posterior distribution (*Geman and Geman, 1984*). To counter autocorrelation in the samples, Neal also endowed the sampler with additional Metropolis–Hastings nudges

to the candidate state parameters between rounds of Gibbs sampling. For these nudges, we use a Gaussian proposal distribution.

In the case of RBME, the state parameters are $\boldsymbol{\theta} = (D, \sigma_{\text{loc}}^2)$. Even with Neal's sampler, a large number of samples are required to estimate the posterior over this 2D space, potentially requiring hours of computational time per dataset.

To make the problem more tractable, we replace this 2D space with a 1D approximation by neglecting the off-diagonal terms in the covariance matrix for RBME (*Equation 12*). With this approximation, *Equation 12* can be rewritten as the log gamma density as

$$\log p_X(X_i|\phi) \propto -S_i e^{-\phi} - L_i \phi \tag{19}$$

where $\phi = \log\left[4(D\Delta t + \sigma_{\text{loc}}^2)\right]$, $S_i$ is the sum of squared 2D jumps in trajectory $i$, and $L_i$ is the number of jumps. Notice that we cannot distinguish the contributions of $D$ and $\sigma_{\text{loc}}^2$ to $\phi$ without measuring $\sigma_{\text{loc}}^2$ by some other method, such as averaging the negative sequential jump covariance across all trajectories in the dataset. This is the price we pay for a tractable DPMM and is the major disadvantage of this model (see, for instance, *Figure 3A*).

The complete Gibbs sampling routine for our DPMM is the following, which is essentially a modified version of Algorithm 8 from *Neal, 2000*:

1. Draw a random sample $\phi^{(0)} = (\phi_1, ..., \phi_{m_0})$ from a uniform distribution on the interval $[\phi_{\min}, \phi_{\max}]$, where the interval is selected to span the parameter space of interest. Each element of the vector $\phi^{(0)}$ represents a candidate 'state.' At each iteration, we will add or remove states from this vector as the sampler explores the posterior.
2. Assign each trajectory $i$ to a state $k \in \{1, ..., m_0\}$ with log probability proportional to $\log p_X(X_i|\phi_k)$. Let this assignment be $Z_i^{(0)}$.
3. For each iteration $t = 1, 2, ...$
   a. For each trajectory $i = 1, 2, ...$, either set $Z_i^{(t)}$ to a state in the current set $\phi^{(t-1)}$ with probability $(N-1)/(\alpha+N-1)$, or create a new state with probability $\alpha/(\alpha+N-1)$.
      i. If setting to an existing state, choose state $k$ with log probability proportional to $\log n_k + \log p_X(X_i|\phi_k)$, where $nk$ is the number of jumps already assigned to state $k$.
      ii. If creating a new state, pick $m_0$ values of $\phi$ from the interval $[\phi_{\min}, \phi_{\max}]$. Among these, accept a particular value $\phi'$ with log probability proportional to $\log p_X(X_i|\phi')$. Add a new state with this parameter to the set of current states $\phi^{(t-1)}$.
   b. For each state $k$, if there are no trajectories currently assigned to it, remove it from consideration. Otherwise add it to $\phi^{(t)}$, the next set of states, and update its parameter according to a Metropolis–Hastings step as follows:
      i. Propose a new $\phi' \sim \mathcal{N}\left(\phi_k^{(t-1)}, \nu^2\right)$ (resampling if $\phi'$ is outside the range $[\phi_{\min}, \phi_{\max}]$).
      ii. Evaluate the likelihood ratio
$$r = \frac{\prod_{i=1}^{N} p_X(X_i|\phi')^{I_{Z_i=k}}}{\prod_{i=1}^{N} p_X(X_i|\phi_k^{(t-1)})^{I_{Z_i=k}}} \frac{\Phi\left(\frac{\phi_{\max}-\phi_k^{(t-1)}}{\nu}\right) - \Phi\left(\frac{\phi_{\min}-\phi_k^{(t-1)}}{\nu}\right)}{\Phi\left(\frac{\phi_{\max}-\phi'}{\nu}\right) - \Phi\left(\frac{\phi_{\min}-\phi'}{\nu}\right)}$$
      iii. Draw $u \sim \text{Uniform}(0, 1)$. If $r < u$, set $\phi_k^{(t)} = \phi'$. Otherwise set $\phi_k^{(t)} = \phi_k^{(t-1)}$.
4. The posterior mean can be estimated by making a histogram of the samples $\phi^{(t)}$ weighted by their occupations $\boldsymbol{n}^{(t)}$, where $n_k^{(t)}$ is the number of jumps assigned to the state with parameter $\phi_k^{(t)}$ at iteration $t$.
5. Finally, we account for defocalization as discussed in 'Defocalization.'

In this algorithm, $\Phi(x)$ is the unit Gaussian CDF and its contribution to $r$ is required to make an unbiased proposal distribution for the Metropolis–Hastings updates given that $\phi$ is confined to the range $[\phi_{\min}, \phi_{\max}]$. $I_{Z_i=k}$ is the indicator function and is 1 if $Z_i = k$ and 0 otherwise.

While the gamma approximation 19 is what makes DPMMs computationally scalable, it also means that in order to disambiguate the contributions of diffusion and localization error to $\phi$ we need to measure localization error by a different method. This is particularly relevant when accounting for defocalization, which relies on knowledge of $D$ independent of $\sigma_{\text{loc}}^2$. In this article, we always use the mean negative covariance between sequential jumps to estimate localization error prior to launching the Gibbs sampler above. However, this means that the DPMM is only as good as our estimate of $\sigma_{\text{loc}}^2$ – and as demonstrated in *Figure 3* and *Figure 3—figure supplement 1*, our estimate of $\sigma_{\text{loc}}^2$ can

be quite noisy with small numbers of trajectories and starts to fail completely when localization error varies a lot between states. SAs, although they require discretizing the parameter space, handle the problem of localization error in a more graceful manner than DPMMs.

## Accounting for defocalization

We use 'defocalization' to refer to the axial movement of fluorescent emitters out of the microscope's focus during an sptPALM acquisition. Because fluorescent emitters move quickly, defocalization is rapid and often limits trajectory length to a few frames. Due to defocalization, the probability to observe a jump from a fast-moving particle is less than that of a slow-moving particle because the jumps of a fast-moving particle are more likely to land outside the microscope's focus.

Defocalization was considered as an experimental avenue to measure diffusion by *Kues and Kubitscheck, 2002*. In the jump histogram modeling frameworks of *Mazza et al., 2012* and *Hansen et al., 2018*, who investigated the effect in detail, it appears as a correction term. The latter two sets of authors evaluated the defocalization probability by treating the microscope's focal volume as a slab with absorbing boundaries and using the solution to the diffusion equation within these boundaries. Because the boundaries for the focal volume are not actually absorbing, both sets of authors then applied a correction term derived from Monte Carlo simulations of regular Brownian motion to 'correct' their correction.

Here, we provide a simpler alternative that is not based on Monte Carlo simulations, enables nonuniform probabilities of detection in the axial detection, and extends to a broader class of diffusion processes than regular Brownian motion. Although the framework can be extended to tracking with gaps, here we consider the case without gaps in tracking (all jumps are strictly between sequential frames).

Let $f(z, t = 0)$ be the initial profile of particles in the axial direction of the microscope, and let $g(z, \Delta t)$ be the Green's function for the diffusion process at this frame interval. For regular Brownian motion, $g(z, \Delta t) = e^{-z^2/4D\Delta t}/\sqrt{4\pi D\Delta t}$. Then the axial probability density for the particle after one frame interval can be obtained by convolving its initial profile with the Green's function:

$$\text{axial profile after 1 frame interval} = f(z, 0) * g(z, \Delta t)$$

To account for defocalization, we multiply this density with an appropriate transmission function. For example, if our focal volume is a slab with depth $\Delta z$, infinite XY extent, and perfect recall at any point inside the slab (i.e., all particles inside the slab are detected and no particles outside are detected), then our transmission function $T$ is

$$T(z) = \begin{cases} 1 & \text{if } z \in \left[-\frac{\Delta z}{2}, \frac{\Delta z}{2}\right] \\ 0 & \text{otherwise} \end{cases}$$

(This is the transmission function considered by *Mazza et al., 2012* and *Hansen et al., 2018*.) The resulting axial profile is

$$f(z, \Delta t) = T(z) \left[f(z, 0) * g(z, \Delta t)\right]$$

To calculate the axial profile after $n$ frame intervals, we repeat this process iteratively:

$$f(z, n\Delta t) = \text{Diffuse}^{(n)}\left[f(z, 0)\right]$$

where $\text{Diffuse}^{(n)}$ denotes $n$ sequential applications of the function

$$\text{Diffuse}\left[f(z)\right] = T(z)\left[f(z) * g(z, \Delta t)\right]$$

This scheme is illustrated in *Figure 3—figure supplement 2A*. The fraction of particles remaining in focus after $n$ frame intervals can be found by integrating this density:

$$\text{fraction defocalized after } n \text{ frames} = \int_{-\infty}^{+\infty} \text{Diffuse}^{(n)}\left[f(z, 0)\right] dz$$

In the SA and DPMM algorithms, we use this method to account for defocalization in the following way. Suppose that $\tau_k$ is the estimated occupation and $D_k$ is the estimated diffusion coefficient for state $k$. Then, we define the corrected state occupations $\tau'$ such that

$$\tau'_k = \frac{\tau_k/\eta_k}{\sum_{j=1}^{k} \tau_j/\eta_j}$$

$$\eta_k = \int_{-\infty}^{+\infty} \text{Diffuse}^{(1)}[\text{f}(z',0)]\,(z)\,\text{d}z$$

(20)

where $\eta_k$ is the probability for a Brownian motion to remain in focus after one frame interval and $\Delta z$ is the focal depth. While defocalization can be incorporated explicitly into the models for SAs or DPMMs, in practice we find it makes little difference if it used as a final postprocessing step after inferring the posterior mean occupations.

To determine the focal depth $\Delta z$, we used the method described in *Hansen et al., 2017*.

## Method availability

We have implemented SAs as a simple tool (https://github.com/alecheckert/saspt, *Heckert, 2022c*), available on the Python Package Index (PyPI) as https://pypi.org/project/saspt/. Documentation is also available at https://saspt.readthedocs.io/en/latest/.

DPMMs for SPT analysis have a publicly accessible implementation at https://github.com/alecheckert/dpsp, (copy archived at swh:1:rev:2f5196e4cae5943a5822be7c4493df50cd564a0c, *Heckert, 2022a*). As a result of the investigation in this article, we recommend that researchers looking to try these methods start with SAs due to their superior performance.

# Acknowledgements

We thank Luke Lavis for generously synthesizing the Janelia Fluor dyes used in these experiments, Thomas Graham for his creative suggestions about the defocalization problem, Ana Robles for the monumental task of keeping our SPT microscopes in working order, and Anders Hansen and Maxime Woringer for insights and advice at the outset of the project. Claudia Cattoglio provided indispensable advice on Western blots. Portions of this work were performed on shared instrumentation at the UC Berkeley Cancer Research Laboratory Molecular Imaging Center (MIC), supported by The Gordon and Betty Moore Foundation. We thank MIC gurus Holly Aaron and Feather Ives for their assistance with shared microscopes. Sanger sequencing was performed at the UC Berkeley DNA Sequencing Facility. This work was supported by NIH grant 1U54CA231641 (XD) and the Howard Hughes Medical Institute (CC34430, RT). AH was supported by the NIH Stem Cell Biological Engineering predoctoral fellowship T32 GM098218. We would like to thank David Schaffer for his support and feedback throughout the project as a program director for the T32 fellowship.

# Additional information

### Funding

| Funder | Grant reference number | Author |
|---|---|---|
| National Institutes of Health | GM098218 | Alec Heckert |
| Howard Hughes Medical Institute | CC34430 | Robert Tjian |
| National Institutes of Health | 1U54CA231641 | Xavier Darzacq |

The funders had no role in study design, data collection and interpretation, or the decision to submit the work for publication.

### Author contributions

Alec Heckert, Conceptualization, Software, Formal analysis, Investigation, Visualization, Methodology, Writing – original draft, Writing – review and editing; Liza Dahal, Validation, Investigation; Robert Tjian,

Resources, Funding acquisition; Xavier Darzacq, Resources, Funding acquisition, Writing – review and editing

### Author ORCIDs
Alec Heckert http://orcid.org/0000-0001-8748-6645
Xavier Darzacq http://orcid.org/0000-0003-2537-8395

### Decision letter and Author response
Decision letter https://doi.org/10.7554/eLife.70169.sa1
Author response https://doi.org/10.7554/eLife.70169.sa2

---

## Additional files

### Supplementary files
• Supplementary file 1. List of primers used in this article.
• Supplementary file 2. List of plasmids used in this article.
• Supplementary file 3. List of Cas9 guide sequences used in this article.
• Transparent reporting form

### Data availability
The state array (SA) method is publicly available as the pip-installable package saspt (https://github.com/alecheckert/saspt, copy archived at swh:1:rev:773292fc245c01ceb8e1f7a7e50b475aa003f00c). The code to generate optical/dynamic simulations in this paper is publicly available as the GitHub repository sptPALMsim (https://github.com/alecheckert/sptpalmsim, copy archived at swh:1:rev:a72f7fff6329813620354d1209d52c654c31f3fc). The DPMM model has a publicly available implementation at https://github.com/alecheckert/dpsp, (copy archived at swh:1:rev:2f5196e4ca-e5943a5822be7c4493df50cd564a0c). All trajectories from SPT experiments used in this paper have been made available as a Dryad dataset (https://doi.org/10.6078/D13H6N). Code used to generate SPT simulations has been organized as a publicly available GitHub repository (https://github.com/alecheckert/strobesim, copy archived at swh:1:rev:ae3bdbf7ccd9740a249a30d9f20deab8ccceb448). Code used to create trajectories from spaSPT movies by detecting and tracking fluorescent emitters, along with a graphic user interface (GUI) for quality control, has been publicly available as a GitHub repository (https://github.com/alecheckert/quot, copy archived at swh:1:rev:1adf7a0574c62f38140f-1dec2d14555bfc03b9a7). Raw gel images are provided as source data for this manuscript.

The following dataset was generated:

| Author(s) | Year | Dataset title | Dataset URL | Database and Identifier |
| --- | --- | --- | --- | --- |
| Heckert A, Dahal L, Tjian R, Darzacq X | 2021 | Dataset from: Recovering mixtures of fast diffusing states from short single particle trajectories | https://doi.org/10.6078/D13H6N | Dryad Digital Repository, 10.6078/D13H6N |

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
