## [Editor Report]

This paper will be of interest to the cellular biologists who perform single-particle tracking experiments and develop new tracking methodologies. The authors investigate a new way of estimating an unknown number of diffusion states from short single-molecule trajectories. Ideas developed in the paper are likely to be used for further algorithm development. The authors give the users access to a repository on GitHub that contains comprehensive code that supports the paper.

---

## [Decision Letter]

**Decision letter after peer review:**

Thank you for submitting your article "Recovering mixtures of fast diffusing states from short single particle trajectories" for consideration by *eLife*. Your article has been reviewed by 3 peer reviewers, one of whom is a member of our Board of Reviewing Editors, and the evaluation has been overseen by a Reviewing Editor and Anna Akhmanova as the Senior Editor. The following individual involved in review of your submission has agreed to reveal their identity: Maarten Paul (Reviewer #3).

Essential revisions:

The authors are supposed to provide a point by point revision addressing the reviewers comments stated in this letter. The main directions for improvements can be summarized briefly as follows:

1) Rewriting or adding more information and explanations about the methodology, so it becomes accessible to a broader range of *eLife* readers.

2) Adding comparison with existing (alternative/similar) techniques mentioned by the reviewer as well as the case with non-Brownian motion.

3) Justifying the importance of the biologically relevant insights (see the reviewer’s comments) to fit the profile of the journal better.

*Reviewer #1 (Recommendations for the authors):*

The paper is well written and all the parts concerning the "regular" Brownian motion, as the authors also mention themselves, are validated with lots of experiments, covering all possible (parameter) dependencies. With the staring assumptions like that (the type of diffusion and the type of switching), it is difficult to ask for more. My concern is still about the applicability of these techniques to the data that contains anomalous diffusion. The typical experiments with H2B or any other protein binding events, where there is a switching between several types of motion, most of the time show that the "slow" components are always related to anomalous diffusion. In the typical cases, there are 2-3 components where only one most likely would correspond to the regular diffusion. With that respect, it would be interesting to know how the methods work (or break) with such data, for example in the simulations with 2-3 state fractional Brownian motions. The weakness that one can imagine is that the proposed techniques can distill the diffusion coefficients, but the bigger problem is that they cannot "split" the trajectories into the parts that correspond to those states with different parameter values. Such split is most of the time of a higher importance, because it allows for computation of residency times and other typical and intuitive parameters. Having the diffusion rates for the anomalous diffusion (which are only the "apparent" diffusion coefficients that do not have physical interpretation as clear as in the case of the regular Brownian motion). Also splitting trajectories in parts, according to different diffusion rates gives a possibility to create a spatial maps inside a cell/nucleus and observe, for example, where more transient binding is occurring. The attempt to present such type of information can be observed in Figure 5 but is it possible to compare these results with "more classical" approach (staring from SpotOn or any other techniques), that splits the trajectories into different parts, and see if the results produced by the proposed methods are unique and cannot be obtained otherwise. Splitting of the tracks with diffusion constants which are well separated (as in the paper) is not a big problem nowadays (see for example M. Arts, "Particle Mobility Analysis Using Deep Learning and the Moment Scaling Spectrum", or A. Vega "Multistep track segmentation and motion classification for transient mobility analysis"). Those already work better than simple MSD analysis that does not keep track on the order of displacements within a track.

*Reviewer #2 (Recommendations for the authors):*

1. Accuracy and precision are two different things. I would recommend that the authors look up the definition of accuracy (i.e. bias) and precision.

2. Introduction paragraph 3, "despite these advances several problems remain". This is very vague and I don't know what the authors tried to address with these advances. Please rephrase.

3. Introduction paragraph 3, "stroboscopic illumination", do the authors mean stroboscopic activation of excitation?

4. " but because camera integration times are never instantaneous, it cannot be removed entirely". Figure 1C should be supported with images from point spread functions (PSF) of real acquired single-molecules and histograms of intensity, background, and PSF width, which are related to the integration time of the camera to make this claim scientifically sound. Also for S1B, it would be easier to plot a 2D grid of pixels with a greyscale indicating the intensity (similar to Figure 1A, but zoomed-in). The paper misses too many details for their argumentation of the varying localization precision that the paper tries to address. This needs to be expanded so that the localization precision simulations match the reality.

5. The authors introduce a new acronym for sptPALM originally introduce in Manley, Nat. Meth, 2008. I don't see a reason for deviating from this.

6. The authors set out to address challenges by citing work from back in 2006 e.g. ref 19, 20. As the authors know a lot has happened since 2006, which should be discussed to describe an appropriate state-of-the-art. An example of this is the work from Linden Nat. Comm. 2017, which should be cited.

7. One of the major points of ref 30 is to be able to process short trajectories. Paragraph 4 suggests something else. Furthermore, a way to incorporate a changing localization precision over the field of view has been studied in the context of single-molecule kinetics Smith Nat. Com. 2019 and should be cited.

8. The paper contains a missing reference in figure S11 please correct it.

9. The authors introduce ref 30, but benchmark against much older ref 32. A comparison to the tracking methodology that was developed in the Elf lab would be useful for the general readership. The code is available on GitHub.

10. The authors make an approximation that is "strictly true when the localization precision is zero". When does this approximation break down, since this is not a valid assumption (e.g. Figure 1C)?

11. How does a user know from the output if they obtain discretization artifacts from SAs?

12. It would be useful if the authors could quantify the error in figure S2 i.e. add a graph with error vs the number of trajectories. Furthermore, it would be useful to see the impact of a broad distribution that realistically models a varying localization precision (see point 1).

13. Figure S6 shows two distinct diffusion states. My impression is that ref 30 would work on this perfectly fine. I recommend the authors to benchmark against this approach. It would be interesting two see a broad distribution of diffusion coefficients where ref 30 would fail. Here also it would be useful if the authors could quantify the error vs the number of trajectories.

14. Wouldn't it be easier to address defocalization using e.g. an astigmatic lens so that the z position can be estimated? Or would the varying localization precision still be a problem? It would be great if the authors could make this point in the discussion.

*Reviewer #3 (Recommendations for the authors):*

– The first paragraph of the results and first figure nicely describes single-molecule data as a mixture of molecules of different diffusive states and how image acquisition biases the results. In the second paragraph the authors present their new model and Bayesian approaches in a technical way. I think it would be useful at this point to explain their Bayesian approach in such a way that is easier to understand for biologists.

– Introduction; Page 3; "The central problem in spaSPT analysis…" I think it would be useful to add here that it is not only problem to recover the underlying set of dynamic states, but also the transitions between those states. Although this is not really the focus of this study I think it is a relevant aspect of single-particle tracking that should be considered.

– Figure 2A: It is difficult for me to understand these schemes. Possibly some additional description in the figure legend explaining the different terms would help.

– Figure 2C and D: It is unclear to me which of the two methods are used in these figures. A heading above the figures could possible clarify this.

– On page 6: paragraph "Performance of DPMMs and SAs on experimental spaSPT" here biological results are written clearly; however I do miss a description on the performance of (DPMMs and SAs) methods on the biological data and what new features are uncovered with their method.

– It seems to me that the SA method is most applicable for biological data as it considers variable localization error depending on the diffusion coefficient of the molecules, whereas the DPMM method work very well in simulations with known localization error, which is unfortunately is not very realistic in cellular experiments. Could the authors directly compare SA and DPMM for their biological dataset (Figure 4A) and discuss possible differences in the results.

– Could the authors indicate, other than possibly providing more accurate results, what new biological insights are revealed with their method, that are not possible to obtain with MSD analysis. Possibly the authors can compare their experimental results (Figure 4 and 5) to MSD and Spot-On analysis in terms of obtained diffusion rates and fraction of different states. It would be useful to know how big the differences would be, compared to these previous methods.

– The authors mention in the discussion that their methods do not work well with non-Brownian motion. In many cases however confined motion types are relevant to describe the motion of proteins in cells, possibly also for the proteins they analyzed. Could the authors discuss in more detail how serious this limitation is, taking into account the types of anomalous diffusion that has been observed for several proteins for example in the cell nucleus.

– Unfortunately the software code from the State Array method was not available at the presented website (https://gitlab.com/alecheckert/saspt/). It is to praise that the authors plan to publish all their code and source data on publication, but it would be nice to have access this software during review. I think it is important to assure that the software is user-friendly and also accessible for biologists.

– If I understand correctly the experiments described in Figure 4B are done with cells expressing the different variants of RARA-HT from an exogenous promotor either transiently or by making use of stable cell lines. It would be useful if the authors indicate in the legend that this is different from the experiment in Figure 4A where they made use of a CRISPR/Cas9 knock-in. Additionally could the authors indicate the number of technical (cells) and biological replicates from these experiments.

– Finally, the paper is written rather technically, requiring at least some knowledge of Bayesian statistics. I do think it would be useful if the paper would be carefully evaluated to be more accessible for a broad audience and avoid technical terms whenever possible.

[Editors’ note: further revisions were suggested prior to acceptance, as described below.]

Thank you for resubmitting your work entitled "Recovering mixtures of fast diffusing states from short single particle trajectories" for further consideration by *eLife*. Your revised article has been evaluated by Anna Akhmanova (Senior Editor) and a Reviewing Editor.

The manuscript has been improved but there are some remaining issues that need to be addressed, as outlined below:

Essential revisions:

All the reviewers agree that the paper is well written and presents a very valuable data analysis technique but also that the paper has a very strong focus on rather complex and methodological developments, which might be far from the expertise of a general *eLife* reader.

We ask the authors to take into account the comments of Reviewer #4 who has several suggestions on how to improve the readability of the paper and also mentions other recent works presenting similar methods for single trajectory characterisation.

*Reviewer #1 (Recommendations for the authors):*

The authors revised the manuscript very elaborately, taking into account all the comments and adding useful supporting explanations and experiments. The paper is now in very good shape, and even though the description of the methodology is improved, it can still be a difficult read for a general reader of *eLife*, but it is difficult to simplify that even more because the underlying theory is indeed quite complex.

*Reviewer #3 (Recommendations for the authors):*

I think the revised manuscript has improved significantly and become a useful paper addressing important aspects of single-molecule tracking with useful novel analysis methods. The additional simulations will help potential users of these methods to assess the appropriate approach to analyze their SPT data.

I found one typo on line 106 varable-> variable

*Reviewer #4 (Recommendations for the authors):*

I find the current manuscript clear and concise, correctly presenting the method developed as well as a variety of examples of use. The text is easily understandable and presents the different concepts and sections in a very ordered way. Also, the extensive number of figures helps to understand the extent of the method and its applicability to different experimental setups. Note that I have no background in Biology, hence my review is focused on the method and its application to simulated and experimental trajectories, and not on the details of the experimental setup (e.g. lines 257-268 and related figures/supplementary material).

My main concern relates to the benchmark of the method, as I miss an objective evaluation of the accuracy of the method. For instance, while the plots presented in Figure 3A, Figure 4A,…etc give a nice visual understanding of the power of SA, they do not allow for a rigorous comparison and evaluation of the method. In that sense, the plots presented in Figure 4 – Supplement2 C and D give a much better understanding of the accuracy of the method. Being this a rather technical paper focusing on a new method, giving a concise numerical metric (e.g. the mean absolute error) may be of interest to the community. It may also help compare objectively with other methods.

Another point which I found hard to understand was if the method was working at the 'single trajectory' level or in an ensemble of trajectories. From what I understand from line 296, the authors can give a prediction for every trajectory separately. I think that is an important and valuable feature, and perhaps should be highlighted earlier in the text. In this sense, it may also be worth pointing out in the text other recent works presenting similar methods for single trajectory characterization. Indeed, while the approach is slightly different, Ref A also proposes the use of Bayesian inference for extracting diffusion properties from trajectories. The use of machine learning has also been prominent recently for this purpose (see Ref C and the references therein) and may be worth adding a comment in the text. Indeed, in the latter reference, there are some approaches to trajectory segmentation, which may complement one of the flaws of the method stated in the text: dealing with transitions between states within a trajectory.

---

## [Author Response]

Essential revisions:The authors are supposed to provide a point by point revision addressing the reviewers comments stated in this letter. The main directions for improvements can be summarized briefly as follows:1) Rewriting or adding more information and explanations about the methodology, so it becomes accessible to a broader range of eLife readers.2) Adding comparison with existing (alternative/similar) techniques mentioned by the reviewer as well as the case with non-Brownian motion.3) Justifying the importance of the biologically relevant insights (see the reviewer’s comments) to fit the profile of the journal better.

We thank the reviewers for their insightful feedback on the content of this manuscript. Through the reviews, we identified numerous revisions that we believe had a strongly positive impact on the quality and clarity of this work. This letter summarizes the experiments and revisions prompted by these reviews.

The Editors identified three Essential Revisions for the manuscript:

1. Rewriting/adding more information about the methodology, to increase its accessibility to a broader range of readers;

2. Adding comparison with existing (alternative/similar) techniques mentioned by the reviewers as well as the case with non-Brownian motion;

3. Justifying the importance of the biologically relevant insights to fit the profile of the journal better.

In addition to these Essential Revisions, Reviewer #2 highlighted limitations of the simulations for localization error and motion blur in this manuscript. We feel that this is a vital point and it led to the design of a new simulation library (sptPALMsim) that we used in several of the other revisions.

As a result, we begin this document by discussing the issue of realistic simulation raised by Reviewer #2. This is followed by a discussion of the Essential Revisions. Finally, we identified several additional areas for improvement based on the reviewer’s comments, which we discuss at the end. Where included, figure numbers refer to the latest version of the paper.

Revision 1: Evaluating state arrays on more realistic sptPALM simulations

One characteristic of the state array (SA) method is that it infers state occupations in the joint space of diffusion coefficient and localization error. Localization error is then marginalized out in post-processing. The original manuscript claimed that this procedure was robust to heterogeneous localization error in the sample. However, Reviewer #2 highlighted an important limitation of the simulations used to support this assertion: while each simulation features a range of localization errors, the localization error for each state is assumed constant. In real experiments, localization error can be distinct for each individual particle due to effects like defocus, motion blur, and the intrinsically stochastic processes of photon emission and detection (“shot noise”).

To address this, we designed a series of more realistic dynamical and optical simulations of the sptPALM experiment. Whereas in the original manuscript we simulated the paths of particles with different types of motion (plus error), for these new simulations we generated images which are subsequently tracked. As a result, the simulations incorporate effects like defocus, shot noise, camera noise, tracking errors, and motion blur that were not present in the original simulations. We have included a new figure (Figure 4-figure supplement 2) that evaluates state arrays on this data, as well as an examples of movies simulated by this method (Video 4, Video 5, and Video 6). The source code to generate the simulations is publicly available as the sptPALMsim package (https://github.com/alecheckert/sptpalmsim). This package includes a library for general-purpose sptPALM simulation as well as scripts that specifically reproduce the data used in this manuscript.

Likewise, Reviewer #2 also highlighted that the claim about motion blur (“because camera integration times are never instantaneous, [motion blur] cannot be removed entirely”) is not substantiated with data from the manuscript proper. They recommended supporting this statement with images from experimentally observed point spread functions at different integration times.

We agree this point is important, and point to several places it has been made before in the literature (for example, Deschout et al. 2012 (doi:10.1002/jbio.201100078) and Linden et al. 2017 (doi:10.1038/ncomms15115) both provide detailed discussion along with experimental point spread functions). Because of this existing body of work, we did not feel that adding experimentally observed point spread functions would add anything to the paper that has not already been stated better.

Instead, to support the investigations here and address Reviewer #2’s comment, we investigated the magnitude of the effect of motion blur on localization by systematically evaluating our detection and localization pipeline on simulated point spread functions with variable pulse width (Figure 1-figure supplement 1). These investigations allowed us to determine that localization error scales with pulse width and is non-negligible for the pulse widths used in this study.

Essential Revision 1 (pt. 1): Methodology for non-specialists

A central point highlighted by the reviewers was the accessibility of the method for non-specialists. The original manuscript was not geared to the diverse range of backgrounds in eLife’s readership. As one example, Reviewer #3 highlighted the use of probabilistic graphical models in Figure 2. PGMs aren’t very useful for readers unfamiliar with them!

To address this issue, we made several major changes to the manuscript. First, we refactored/replaced Figure 2 with two new figures (now, Figure 2 and Figure 3). The first new figure compares the models in this manuscript with an intuitive, visual presentation geared toward non-specialists. The second new figure dives deeper into state arrays and DPMMs as applied to regular Brownian motion, and contains most of the information formerly in Figure 2. The graphical models from Figure 2 have been moved to their own figure (Figure 2-figure supplement 1). We believe this increases the intelligibility of the method for non-specialists, while retaining the original information for those versed in probabilistic models.

Additionally, we improved the exposition of methodology. This included two parts. First, we rewrote the first Results section describing the methods with an eye to eLife’s readership. The new version introduces the concept of a mixture model at a basic level, and follows by modifying this model to yield state arrays and DPMMs. We also devote more space to discussing how the design of state arrays and DPMMs is intended to address the fundamental problem of estimating a discrete or continuous dynamic profile.

The second part of our revision was an overhaul of the Methods. The new version is essentially a more detailed version of the first Results section, beginning with classic Bayesian finite state mixtures and deriving DPMMs and state arrays as an extension and special case, respectively. We have taken several steps to increase the accessibility of this material to non-specialists, including (1) removing unnecessary discussion of specialist topics, (2) breaking out long derivations to separate sections, and (3) algorithmic (step-by-step) perspectives of the different classes of mixture models discussed in the paper.

Essential Revision 1 (pt. 2): Method accessibility

The original manuscript linked to a GitLab repository implementing the methods described in the manuscript. That repo was fairly informal and suffered from sparse documentation, absence of a comprehensive testing suite, and some permissions issues identified by Reviewer #3.

To improve the accessibility of the method, we have reimplemented state arrays as the publicly available, pip-installable Python package saspt. This software includes a full testing suite, a flexible object API, and a comprehensive set of documentation (accessible here: https://saspt.readthedocs.io/en/latest/) that includes a guide for getting started running state arrays. This documentation has led to useful feedback from the current users of the method. We have linked to this package in the new version of the manuscript.

Essential Revision 2: Comparison with existing techniques

The state-of-the-art in finite-state sptPALM analysis is represented by the vbSPT tool (Persson et al. Nature Methods 2013). Reviewer #2 suggested a comparison between the state array method and vbSPT.

To compare vbSPT and state arrays, we performed two experiments using simulated sptPALM movies produced with the sptPALMsim package. First, we simulated a range of two-state Brownian models with variable diffusion coefficients and state occupations. In these comparisons, the simulated sptPALM movie was tracked with our usual pipeline (Localization and tracking, Methods), then the resulting trajectories were passed as input to both the state array and vbSPT methods. The results are summarized in a new figure (Figure 4-figure supplement 6).

Comparing the two methods is not entirely trivial because (1) vbSPT and state arrays have different outputs (vbSPT returns a variable number of discrete states, while state arrays return a distribution over an array of parameters), and (2) vbSPT frequently returned more states than the true number of simulated states. Consequently, we chose the following method for this comparison:

– To compare accuracy of diffusion coefficient retrieval: For state arrays, we took the diffusion coefficient corresponding to the fastest mode in the posterior distribution. For vbSPT, we took the state with the diffusion coefficient closest to the true diffusion coefficient.

– To compare accuracy of state occupation retrieval, we integrated the occupations of all states above or below 1.0 µm2 s-1 (for both methods).

The result was close agreement between the vbSPT and state array methods, both of which accurately recovered state occupation and diffusion coefficient for the two-state model (Figure 4-figure supplement 6B). We concluded that vbSPT and state arrays had similar performance for this simple model. We did not assess the accuracy of the transition rate matrix estimated by vbSPT (no transitions were allowed in this experiment).

For our second comparison, we compared the output of vbSPT and state arrays on 20 dynamic models of increasing complexity (Figure 4-figure supplement 6C). This included situations that we consider extremely challenging for any current method (including both state arrays and vbSPT), such as models with 10 diffusing states and “clusters” of diffusing states with similar state parameters. In this comparison, we compared the ability of vbSPT and state arrays to recover the number of diffusing states. We found that while both methods performed well on simple models, they were prone to distinct kinds of errors for more complex models (vbSPT tended to overestimate the number of states, while state arrays tended to underestimate). Additionally, vbSPT inaccurately retrieved the diffusion coefficients for slow-moving states, likely due to its treatment of localization error. We concluded that state arrays and vbSPT are likely complementary methods, with state arrays being more applicable to complex non-discrete dynamic profiles and vbSPT more appropriate to recover the transition rates for simple models.

Essential Revision: Non-Brownian motion.

Reviewers #1 and #3 inquired about the applicability of these methods to non-Brownian motion. Because the state array method is applicable to any kind of probabilistic motion model it extends naturally to non-Brownian motion, although the inference routine is more expensive because a higher-dimensional parameter grid is necessary.

To address the reviewers’ comments we implemented a state array for fractional Brownian motion (FBM), a popular anomalous diffusion model that generalizes regular Brownian motion to allow for subdiffusion and superdiffusion. In our version of this model, there are three parameters:

– A scaling coefficient parametrizing the magnitude of the jumps, similar to the diffusion coefficient for regular Brownian motion;

– The Hurst parameter, parametrizing temporal correlations in the jumps;

– Localization error variance.

To distinguish this modified model from the original FBM as introduced by Mandelbrot & Van Ness SIAM Review 1968, we refer to it as “fractional Brownian motion with localization error” (FBME).

The state array for FBME is a 3-dimensional array over scaling coefficient, Hurst parameter, and localization error variance. As with the RBME state array, we marginalize out the localization error in post-processing. This yields a two-dimensional function of scaling coefficient and Hurst parameter (rather than the one-dimensional function over diffusion coefficient afforded by the RBME state array). We then proceeded to evaluate this state array’s ability to recover FBM model parameters from sptPALM movies simulated by the sptPALMsim package.

The results of these investigations are summarized in Figure 4-figure supplement 7 and Figure 4-figure supplement 8, with an accompanying movie (Video 6). With short pulse widths, this state array accurately recovered mixtures of FBM states (both subdiffusive and superdiffusive) (Figure 4-figure supplement 7). However, as we increased pulse width, we found that the state array systematically underestimated the scaling coefficient at low Hurst parameters (Figure 4-figure supplement 8A). This effect appears to be due to motion blur averaging out the high-frequency motion that characterizes FBM at low Hurst parameters (Figure 4-figure supplement 8B). This interaction between FBM and motion blur is unexpected and, as far as we are aware, previously unreported.

The state array used for this investigation is publicly available in the saspt package by passing the option likelihood_type = ‘fbme’.

Essential Revision 3: Biological insights

Reviewer #1 and #3 asked about the novel biological insights offered by these methods. These comments made us realize that the previous version of the manuscript included essentially no discussion of insights gleaned from the application of state arrays to real sptPALM data. While the purpose of the manuscript is to compare methods rather than investigate novel biology, we agree that this addition would help eLife’s readership understand better where these methods could be applicable.

To address this, we introduced a section in the discussion that focuses on the insights from these methods. We highlight that the dynamic profile for RARA-HaloTag unexpectedly displayed a non-discrete spectrum of diffusing states between 0.1 and 10.0 µm2 s-1, in contrast the more discrete profiles of HaloTag, HaloTag-NLS, or H2B-HaloTag. We feel that this serves as a demonstration of the ability of state arrays to approximate non-discrete profiles, which sets it apart from existing methods.

Here, we discuss other points raised by the reviewers that led to improvements in the manuscript.

1. In the original manuscript, we used the acronym “spaSPT” (stroboscopic photoactivated single particle tracking) to refer to the tracking experiment, following Hansen and Woringer et al., *eLife* 2018. Reviewer #2 felt that the acronym “sptPALM” (single particle tracking photoactivated localization microscopy), introduced by Manley et al., *Nature Methods* 2008, is preferable. We have no intrinsic attachment to either so we have changed all instances of “spaSPT” to “sptPALM” in the manuscript.

2. Reviewer #2 suggested a more explicit discussion of where the DPMM method breaks down, particularly where localization error is concerned. We had intended Figure 4 and its associated supplements to show that the DPMM breaks down when localization error is highly heterogeneous in the sample; this is the method’s major failing point (and the main reason why it is outperformed by state arrays). We have attempted to make this clearer in the text referring to these figures.

3. Reviewer #2 proposed quantifying the error in state occupation estimates as a function of the number of trajectories used. This is the main purpose of Figure 4—figure supplement 3, so we have attempted to make the reference to this figure clearer in the main text.

4. The original manuscript did not recommend any way to determine when and where discretization artefacts occur when using state arrays. We agree with Reviewer #2 that this would be a helpful addition. We have added a note about it in the Discussion. In this note, we also indicate that the saspt package includes “out-of-the-box” state arrays that have been selected to avoid discretization artefacts based on our own experiments.

5. Reviewer #3 included numerous helpful comments on figure clarity, including that (i) the original Figure 2C and Figure 2D did not indicate which method (DPMMs or SAs) were used, (ii) Figure 5B did not make it clear that which constructs were expressed from an exogenous promoter, (iii) the crosshairs in the original Figure 2D were difficult to see, (iv) Figure 5A-C does not state which method was used except in its caption, and (v) confusing labeling in Figure 6B. We have modified these figures in response to these comments by (i) adding a legend for Figure 2C/2D, (ii) marking the for constructs expressed from an exogenous promoter to Figure 5B, (iii) changing the color of the crosshairs in Figure 2D, (iv) stating that Figure 5A-C uses state arrays in the figure itself, and (v) rearranging Figure 6B so that each subpopulation is clearly connected to the original state array posterior distribution by an arrow.

6. Introducing point spread function (PSF) astigmatism via a cylindrical lens is a common way to estimate the z-position of emitters in fixed-cell PALM and STORM. Reviewer #2 suggested a discussion of the applicability of this method to tracking in the Introduction, and were especially interested in whether it could be used to address the defocalization problem. We have included two additional comments in the introduction, which are (1) astigmatism does not solve the issue of finite focal depth, since the focal depth with a cylindrical lens is still smaller than the depth of a typical eukaryotic cell, and (2) while astigmatism could potentially be a powerful tool for resolving three-dimensional motion in tracking, currently there do not exist subpixel localization methods that can effectively distinguish astigmatism from motion blur. (sptPALM differs from fixed-cell PALM and STORM methods in that the fixed-cell methods work with much higher numbers of photons and do not contend with motion blur.)

7. Reviewers #1 and #3 were both interested in the calculation of transition rates between diffusive states. They highlighted that although the methods presented do not infer transition rates, a more detailed discussion of this limitation would be appreciated. In addition to the existing Figures S4E and S7 (which compare the performance of SAs/DPMMs on simulations with transitions between diffusive states), we have added a discussion in the Introduction on this limitation of the state array method. This change makes it clearer that the methods investigated in this manuscript do *not* infer transitions between states, in contrast to methods like vbSPT.

8. Reviewer #2 identified ambiguity in the text between the usage of the words “accuracy” and “precision”. We have revised several parts of the manuscript to better elucidate where we discuss the accuracy (systematic deviations from the true underlying value) and precision (variation about the true value) of the inference methods.

9. Reviewer #2 raised a point about a potentially confusing part of the text: while the manuscript states that DPMMs and SAs deal with the issue of an unknown number of diffusive states, it appeared to this reviewer that “it is not possible to classify trajectory segments into a state”. We apologize for any ambiguity and wish to highlight that SAs do provide a posterior probability distribution over states for each trajectory; assignment of a trajectory to a given state can be accomplished through either the max a posteriori or mean posterior estimates. We have attempted to make this point clearer in the discussion of the method, and thank Reviewer #2 for pointing out this ambiguity.

10. Reviewer #2 highlighted a vague sentence in the introductory paragraph (“Despite these advances, several problems remain…”). We recognize that some of the confusion comes from the phrase “Despite these advances”, which was intended to refer to the contents of the previous paragraph (“Advances in the past two decades…”). We have removed this phrase and rephrased this paragraph and the following two for clarity. The intent of these paragraphs is to discuss current challenges with the sptPALM experiment.

11. Reviewer #2 pointed out that “stroboscopic illumination” is unclear. We have added a clarifying comment that this refers to pulses of the excitation laser, as shown in Figure 1A. In addition, Figure 1—figure supplement 1 provides additional illustrations of the motivation for stroboscopic illumination by investigating the effect of motion blur on localization precision in simulated SPT videos.

12. Reviewer #2 suggested rephrasing the discussion of reference #30 (Persson et al., 2013), particularly its applicability to short trajectories. We have highlighted explicitly that this method works with short trajectories in the Introduction. Additionally, the new comparisons on simulated sptPALM in Figure 4—figure supplement 6 show clearly that vbSPT accurately recovers state occupations and diffusion coefficients for simple models, given short trajectories.

[Editors' note: further revisions were suggested prior to acceptance, as described below.]

Reviewer #4 (Recommendations for the authors):I find the current manuscript clear and concise, correctly presenting the method developed as well as a variety of examples of use. The text is easily understandable and presents the different concepts and sections in a very ordered way. Also, the extensive number of figures helps to understand the extent of the method and its applicability to different experimental setups. Note that I have no background in Biology, hence my review is focused on the method and its application to simulated and experimental trajectories, and not on the details of the experimental setup (e.g. lines 257-268 and related figures/supplementary material).My main concern relates to the benchmark of the method, as I miss an objective evaluation of the accuracy of the method. For instance, while the plots presented in Figure 3A, Figure 4A,…etc give a nice visual understanding of the power of SA, they do not allow for a rigorous comparison and evaluation of the method. In that sense, the plots presented in Figure 4 – Supplement2 C and D give a much better understanding of the accuracy of the method. Being this a rather technical paper focusing on a new method, giving a concise numerical metric (e.g. the mean absolute error) may be of interest to the community. It may also help compare objectively with other methods.Another point which I found hard to understand was if the method was working at the 'single trajectory' level or in an ensemble of trajectories. From what I understand from line 296, the authors can give a prediction for every trajectory separately. I think that is an important and valuable feature, and perhaps should be highlighted earlier in the text. In this sense, it may also be worth pointing out in the text other recent works presenting similar methods for single trajectory characterization. Indeed, while the approach is slightly different, Ref A also proposes the use of Bayesian inference for extracting diffusion properties from trajectories. The use of machine learning has also been prominent recently for this purpose (see Ref C and the references therein) and may be worth adding a comment in the text. Indeed, in the latter reference, there are some approaches to trajectory segmentation, which may complement one of the flaws of the method stated in the text: dealing with transitions between states within a trajectory.

A central part of the manuscript compares the ability of three methods – MSD histograms, Dirichlet process mixture models (DPMMs), and state arrays (SAs) – to recover state occupations on several kinds of simulated sptPALM datasets. While this comparison was intended to evaluate the performance of each method on data with known ground truth, the comparisons were mostly visual. Most of the original figures showed the dynamic profiles obtained from each of the three methods as line plots.

Reviewer #4's primary critique was a lack of clear objective, numerical benchmarks to ground these visual presentations. This issue was compounded by the lack of y-axis labels for many plots. Reviewer #4 suggested a concise numerical metric, such as mean absolute error (MAE), would be useful for readers. They also identified several related changes that could improve the clarity and improve the quantitative rigor of these comparisons.

After reviewing the manuscript, we agree with Reviewer #4 that this was a clear gap in the existing manuscript. We also feel that it would be helpful to have a single table of numerical accuracies for each method across a broad range of simulations (see note on the new Figure 4—figure supplement 2, below). As a result, we revised most of the figures that focus on method comparison.

First, in figures evaluating the accuracy of state occupation estimates, we have introduced tables of mean absolute errors for each of the methods being compared:

– Figure 4 has been expanded to include the MAE in occupation estimates for each of the three methods when evaluated on a 3-state model (Figure 4E), as shown visually in Figure 4D.

– Figure 4—figure supplement 3 has been split into two figures (Figure 4—figure supplement 3 and Figure 4—figure supplement 4), each of which now includes a table of numerical accuracies (MAE) evaluated against 2- or 4-state models, respectively. This also provides a quantitative comparison of how each methods' accuracy improves as a function of sample size, a point that was only made qualitatively in the previous version of the manuscript. We have also attempted to show the distribution of MAEs for each replicate of these simulations individually (Figure 4—figure supplement 3C and Figure 4—figure supplement 4C).

– Figure 4—figure supplement 5 has been split into two figures (Figure 4—figure supplement 8 and Figure 4—figure supplement 9) that each include tables of mean absolute error on their respective simulations. These is particularly important, given Reviewer #4's comments on Figure 4—figure supplement 5A.

Second, we have introduced a new figure (Figure 4—figure supplement 2) that focuses on numerical comparisons of accuracy between the three methods. This figure takes the simulations from Figure 4—figure supplement 1 (across six kinds of dynamic models) and evaluates a quantitative metric of error, summarizing the results as a table. Because these simulations include both discrete and continuous dynamic profiles, we opted to quantify accuracy as the root mean squared deviation of the estimated CDF from the ground truth CDF, as shown in Figure 4—figure supplement 2A. (This choice of error metric can be used, for instance, on the log uniform densities that cannot be separated into discrete states.) These experiments provide a quantitative demonstration of DPMM's inaccuracy when the localization error is variable. We also noted from these figures that even on ideal sptPALM data, an error rate of 0.01 to 0.05 CDF RMSD is to be expected for SAs.

Third, in Figure 4—figure supplement 10 (which compares state arrays against vbSPT), we have included a column for the error in the estimated number of states relative to the true number of states for each method.

As a result of these changes, the new version of the manuscript places more focus on quantitative comparisons of the three techniques and provides tables of accuracies that benchmark the performance improvements afforded by state arrays. We also feel they serve as a better illustration of the limits of state arrays – for instance, we rarely see the MAE in state occupation estimates drop below 1%, a point we make in the revised text.